# Attention, Please! Revisiting Attentive Probing through the lens of Efficiency

**Bill Psomas**[1][*]    **Dionysis Christopoulos**[2][*]    **Eirini Baltzi**[2]    **Ioannis Kakogeorgiou**[7]
**Tilemachos Aravanis**[1]    **Nikos Komodakis**[3,4,5]    **Konstantinos Karantzalos**[2]
**Yannis Avrithis**[6]    **Giorgos Tolias**[1]

[1]VRG, FEE, Czech Technical University in Prague    [2]National Technical University of Athens
[3]University of Crete    [4]Archimedes, Athena RC    [5]IACM-FORTH    [6]IARAI
[7]IIT, NCSR "Demokritos"

## Abstract

As fine-tuning becomes impractical at scale, probing is emerging as the preferred evaluation protocol. However, standard linear probing can understate the capability of models whose pre-training optimizes local representations rather than an explicit global representation. This motivates attentive probing, an alternative that uses attention to selectively aggregate patch-level features. Despite growing adoption, attentive probing is still underexplored: existing approaches are often over-parameterized and computationally inefficient. In this work, we revisit attentive probing through the lens of the accuracy vs. parameter-efficiency trade-off. We present the first comprehensive study of existing methods, analyzing their design choices and benchmarking their performance. Building on these insights, we propose efficient probing (EP), a lightweight yet effective multi-query cross-attention mechanism that eliminates redundant projections and reduces the number of trainable parameters. Across multiple benchmarks and pre-training paradigms, EP consistently outperforms linear probing and previous attentive probing methods, and remains effective when combined with parameter-efficient fine-tuning. Beyond evaluation, our analysis uncovers emerging properties of EP, including complementary attention maps, which open new directions for leveraging probing beyond protocol design. Project page: https://vrg.fel.cvut.cz/ep/.

## 1 Introduction

The past few years have witnessed remarkable progress in *representation learning*, with pre-training paradigms ranging from self-supervised learning (Chen et al., 2020; Caron et al., 2021; Kakogeorgiou et al., 2022), to vision–language models (Radford et al., 2021; Jia et al., 2021), and autoregressive architectures (El-Nouby et al., 2024; Fini et al., 2025). These diverse approaches share a *common goal*: learning rich, transferable visual representations that minimize reliance on task-specific labels and scale to large datasets. Evaluating the quality of such pre-trained representations is therefore *central* to measuring progress. Conventional evaluation protocols include $k$-NN classification, linear probing (LP), and full fine-tuning (FT). While $k$-NN and LP assess the quality of the learned representations under a *frozen* backbone, FT measures the utility of pre-training as *initialization* for downstream tasks. Although FT achieves the highest performance, it is increasingly viewed as unsustainable and prohibitive at scale (Xin et al., 2024; Shuttleworth et al., 2024; Zou et al., 2023). As a result, probing is emerging as a practical evaluation protocol (Oquab et al., 2024; Bardes et al., 2024; Darcet et al., 2025).

However, probing protocols remain *misaligned* with many pre-training approaches. Standard LP typically relies on a single *global* representation. e.g., the `[CLS]`, which is well-suited for architectures trained with global objectives, but poorly reflects the potential of models such as masked image modeling (He et al., 2022), autoregressive (El-Nouby et al., 2024), or diffusion (Ma et al., 2024), where valuable information is distributed across *local* representations. This gap motivates the rise of *attentive probing* (Darcet et al., 2025; Chen et al., 2023; Bardes et al., 2024). Despite its promise, attentive probing remains underexplored. Existing methods vary significantly in design,

---

[*]Equal contribution

often suffering from *excessive parameterization* and *inefficiency*. Moreover, the connection between how attention aggregates features and why it improves predictive performance remains unclear.

In this work, we address these limitations by conducting the first comprehensive study of attentive probing, revisiting its design through the lens of the *accuracy vs. parameter-efficiency trade-off*. We introduce a unified framework that encompasses a wide range of attention-based aggregation methods—including those proposed for probing (El-Nouby et al., 2024; Bardes et al., 2024; Darcet et al., 2025) and others from unrelated tasks (Psomas et al., 2023; Noh et al., 2017; Rymarczyk et al., 2021). Through this framework, we derive *efficient probing* (EP), a simple multi-query cross-attention mechanism that eliminates redundant projections, reduces parameter count and computational cost, while matching or surpassing prior state-of-the-art performance. Moving beyond the parameter-efficient probing (PEP), we compare EP against parameter-efficient fine-tuning (PEFT) methods. We find that EP remains beneficial even when PEFT is allowed: combining EP with LoRA (Hu et al., 2022) yields configurations that dominate both pure probing and pure LoRA.

Beyond efficiency and accuracy, a common ingredient of both existing methods and EP is the use of multiple independent *attention predictors* (e.g., heads or queries). We show that a predictor's contribution to classification accuracy correlates with its localization quality: predictors with sharper, foreground-focused attention drive larger accuracy gains. Rather than resorting to shortcut learning, e.g., leveraging background cues like water to classify a "fish", EP's predictors consistently attend to the object, improving interpretability, robustness, and performance. Notably, the attention maps of EP are more diverse and complementary than those of existing methods, with different predictors specializing in distinct object regions. Our contributions are threefold:

1. We conduct the first systematic benchmark and analysis of attentive probing methods across diverse pre-training paradigms, comparing their accuracy, efficiency, and design choices.
2. We introduce *efficient probing* (EP), which achieves state-of-the-art accuracy, while bringing substantial gains in compute, memory, and parameter efficiency.
3. We uncover a correlation between spatial localization and predictive performance, and show that EP produces diverse, complementary, and interpretable attention maps.

## 2 RELATED WORK

Self-supervised learning (SSL) has transformed visual representation learning, with evaluation typically performed via (i) $k$-$NN$ on frozen features, (ii) *linear probing* (LP) using a shallow classifier on a frozen encoder, or (iii) *fine-tuning* (FT) the entire model. Although FT achieves the highest accuracy, it is *computationally expensive*, motivating the evaluation under frozen backbones. Two dominant SSL paradigms are joint embedding architectures (JEA) and masked image modeling (MIM). JEA methods (e.g., DINO (Caron et al., 2021)) contrast or cluster augmentations to learn global representations via a `[CLS]` token or pooled features. In contrast, MIM methods (e.g., MAE (He et al., 2022)) reconstruct masked regions, yielding localized, patch-distributed representations. This global vs. local distinction affects evaluation: LP is effective for JEA (Caron et al., 2021) but under-performs for MIM (He et al., 2022; Przewikeźlikowski et al., 2025), where discriminative information is not concentrated in a single token.

Beyond SSL, *vision–language models* (VLMs) pre-train on web-scale image–text corpora (e.g., LAION (Schuhmann et al., 2022)) and optimize cross-modal alignment (e.g., CLIP (Ilharco et al., 2021; Radford et al., 2021)). Although these models expose strong global descriptors, much of the signal remains distributed across patch tokens, making attentive aggregation appealing at probe time. Likewise, *auto-regressive* (AR) families (e.g., AIM/AIMv2 (El-Nouby et al., 2024; Fini et al., 2025)) and *diffusion-based* transformers (e.g., DiT (Peebles & Xie, 2023)) are primarily trained for generation (next-token prediction or denoising), not representation learning per se; nevertheless, their frozen features can be probed to assess representation quality. In all these cases, protocols that assume a single discriminative token may under-utilize the information spread across patches.

To address this, recent work explores *attentive probing* (El-Nouby et al., 2024; Chen et al., 2023; Bardes et al., 2024; Darcet et al., 2025), which learns attention to selectively aggregate patch tokens into informative descriptors for a linear classifier. While methods like AIM (El-Nouby et al., 2024), CAE (Chen et al., 2023), and V-JEPA (Bardes et al., 2024) adopt this idea, no unified evaluation exists. We fill this gap with a comprehensive benchmark and introduce a novel attention mechanism achieving a strong accuracy–efficiency trade-off. Additional related work appears in Appendix A.

## 3 METHOD

### 3.1 ATTENTIVE POOLING

**Preliminaries.** Let $X \in \mathbb{R}^{D_i \times N}$ be the *feature matrix* obtained from a *pre-trained* and *frozen* ViT backbone, where $D_i$ is the number of feature channels and $N = W \times H$ the number of features, one per image patch across the spatial dimensions $W \times H$. Given the *input features* $X$, the goal is to generate an *output image-level feature* $\boldsymbol{y} \in \mathbb{R}^{D_o}$ by applying an *attentive pooling* mechanism. The output feature is used to train a $C$-way linear classifier with $D_o(C + 1)$ parameters.

We consider $M$ *attention predictors*, to be discussed in subsection 3.2. For each predictor $j \in \{1, \ldots, M\}$, let $\boldsymbol{a}_j \in \mathbb{R}^N$ be the $\ell_1$-normalized *attention vector* it generates. Each vector, reshaped to $W \times H$, is an attention map indicating the locations on which the predictor focuses. Let $V \in \mathbb{R}^{D_o \times N}$ be the *value features*, commonly obtained by a linear transformation $V = W_V X$, where $W_V \in \mathbb{R}^{D_o \times D_i}$ is a learnable *projection matrix*.

Let the output feature $\boldsymbol{y}$, value features $V$ and projection matrix $W_V$ be partitioned into $M$ subvectors / submatrices according to

$$\boldsymbol{y} = \begin{bmatrix} \boldsymbol{y}_1 \\ \vdots \\ \boldsymbol{y}_M \end{bmatrix}, V = \begin{bmatrix} V_1 \\ \vdots \\ V_M \end{bmatrix}, W_V = \begin{bmatrix} W_{V_1} \\ \vdots \\ W_{V_M} \end{bmatrix}, \tag{1}$$

with $\boldsymbol{y}_j \in \mathbb{R}^{d_o}$, $V_j \in \mathbb{R}^{d_o \times N}$, $W_{V_j} \in \mathbb{R}^{d_o \times D_i}$ and $d_o = \frac{D_o}{M}$.

The attentive pooling operation is then given by

$$\boldsymbol{y}_j = V_j \boldsymbol{a}_j = W_{V_j} X \boldsymbol{a}_j. \tag{2}$$

Each attention predictor is responsible for the weighted pooling of $N$ features into a $d_o$-dimensional subspace of the final representation space. In the following, we explore existing and novel ways for designing these attention predictors. We focus on the number of additional parameters to be learnt on top of the frozen backbone and the computational complexity of the pooling operation.

### 3.2 ATTENTION PREDICTORS

**Multi-Head Cross-Attention (MHCA).** A vanilla approach is to perform multi-head cross-attention between the input features and an *input vector* $\boldsymbol{u} \in \mathbb{R}^{D_i}$, where each head corresponds to a separate attention predictor. The *query feature* $\boldsymbol{q} \in \mathbb{R}^{D_a}$ and *key features* $K \in \mathbb{R}^{D_a \times N}$ are obtained by linear transformations $\boldsymbol{q} = W_Q \boldsymbol{u}$, $K = W_K X$ with projection matrices $W_Q, W_K \in \mathbb{R}^{D_a \times D_i}$.

Let the query feature $\boldsymbol{q}$ and projection matrix $W_Q$ be partitioned into $M$ subvectors / submatrices according to

$$\boldsymbol{q} = \begin{bmatrix} \boldsymbol{q}_1 \\ \vdots \\ \boldsymbol{q}_M \end{bmatrix}, W_Q = \begin{bmatrix} W_{Q_1} \\ \vdots \\ W_{Q_M} \end{bmatrix}, \tag{3}$$

with $\boldsymbol{q}_j = W_{Q_j} \boldsymbol{u} \in \mathbb{R}^{d_a}$, $W_{Q_j} \in \mathbb{R}^{d_a \times D_i}$ and $d_a = \frac{D_a}{M}$. Similarly, let the key features $K$ and projection matrix $W_K$ be partitioned according to

$$K = \begin{bmatrix} K_1 \\ \vdots \\ K_M \end{bmatrix}, W_K = \begin{bmatrix} W_{K_1} \\ \vdots \\ W_{K_M} \end{bmatrix}, \tag{4}$$

with $K_j = W_{K_j} X \in \mathbb{R}^{d_a \times N}$ and $W_{K_j} \in \mathbb{R}^{d_a \times D_i}$.

The attention vector for head $j$ is then given by

$$\boldsymbol{a}_j = \text{softmax}(\hat{\boldsymbol{a}}_j) \tag{5}$$

with

$$\hat{\boldsymbol{a}}_j = K_j^\top \boldsymbol{q}_j = (W_{K_j} X)^\top (W_{Q_j} \boldsymbol{u}). \tag{6}$$

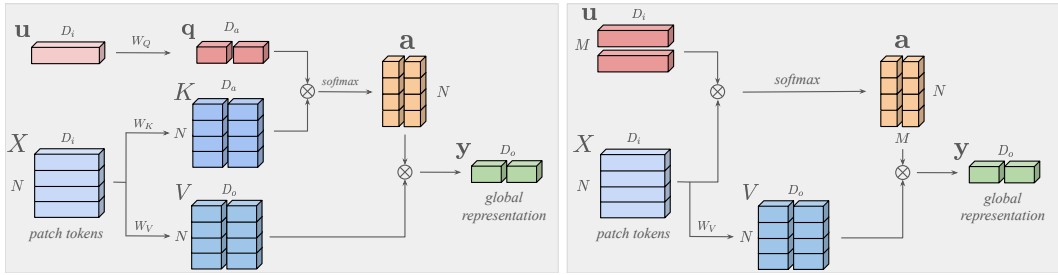

Figure 1: *Comparison of multi-head cross-attention (MHCA, left) vs. our multi-query cross-attention (EP, right).* MHCA uses an input vector $\boldsymbol{u}$ projected into query space and interacts with key features $K$ in (two) separate subspaces, each corresponding to an attention predictor. Attention predictor outputs $\boldsymbol{a}_j$ are used to aggregate value features $V$ into sub-vectors $\boldsymbol{y}_j$, forming the final output $\boldsymbol{y}$. In contrast, EP employs (two) *learnable queries* $\boldsymbol{q}_j$, one per attention predictor, to compute attention with input features directly in the full representation space. Attention predictor outputs $\boldsymbol{a}_j$ are used as in MHCA to perform the aggregation.

That is, the input features $X$ and input vector $\boldsymbol{u}$ are projected to $d_a$-dimensional subspaces where attention subvectors are computed via dot product followed by softmax normalization over patches. This attention predictor requires $D_a(2D_i + 1)$ parameters and has complexity $\mathcal{O}(ND_aD_i)$. As discussed in subsection 3.3, there are several existing methods that fit within this generic framework.

**MHCA with a learnable query.** If we consider input vector $\boldsymbol{u}$ to be learnable, then there is no need for the projection matrix $W_Q$ in (6). Instead, we can set the query feature $\boldsymbol{q}$ to be learnable, thus absorbing $W_Q$ and $\boldsymbol{u}$:

$$\hat{\boldsymbol{a}}_j = (W_{K_j}X)^\top \boldsymbol{q}_j = X^\top W_{K_j}^\top \boldsymbol{q}_j \tag{7}$$

where the query feature $\boldsymbol{q}_j \in \mathbb{R}^{d_a}$ is learnable.

We observe that $W_{K_j}^\top$ maps $\boldsymbol{q}_j$ to the $D_i$-dimensional space of input features to compute the attention vector. Thus, standard MHCA ensures that each query subvector is interacting with the full representation space of the input features, despite being defined in a smaller dimensional space. Using a learnable query feature directly simplifies the architecture, reduces the amount of computations and the number of parameters to $D_a(D_i + 1)$.

In order to explore ways for reducing the total number of parameters and to better understand the role of key transformation, we simplify the architecture by removing it. Letting $W_K$ be fixed to the identity matrix, (7) becomes

$$\hat{\boldsymbol{a}}_j = X_j^\top \boldsymbol{q}_j \tag{8}$$

where the feature matrix $X$ is partitioned into $M$ submatrices according to

$$X = \begin{bmatrix} X_1 \\ \vdots \\ X_M \end{bmatrix}, \tag{9}$$

with $X_j \in \mathbb{R}^{d_i \times N}$ and $d_i = \frac{D_i}{M}$. We thus observe that the query feature only interacts with a $d_i$-dimensional subspace of the input features for $M > 1$, which is a *limitation*. We experimentally verify that setting $W_K$ to identity matrix results in a noticeable performance drop. Based on this observation, in the following, we revisit (7) and design attention predictors that have less parameters and require less compute but do not experience a performance drop due to mathematical equivalence.

**Parameter-efficient Multi-Query Cross-Attention (MQCA).** Instead of using the key submatrices $W_{K_j}$ to project the query subvectors $\boldsymbol{q}_j$ to the $D_i$-dimensional input feature space, we propose to learn $M$ effective query features $\boldsymbol{u}_j := W_{K_j}^\top \boldsymbol{q}_j \in \mathbb{R}^{D_i}$ in that space directly (Figure 1). Thus, $\boldsymbol{u}_j$ absorbs $W_{K_j}$ and $\boldsymbol{q}_j$ and attention prediction becomes

$$\hat{\boldsymbol{a}}_j = X^\top \boldsymbol{u}_j \tag{10}$$

for $j \in \{1, \dots, M\}$. As a result, there are no projection matrices and there are no parameters other than the learnable query features $\boldsymbol{u}_j$.

This choice reduces the number of additional parameters to be learned and saves from one more matrix-vector multiplication. In particular, it requires $D_i M$ parameters for the attention compared to $D_a(D_i + 1)$ for (6), while the number of operations drops to $ND_i M$ compared to $ND_a(D_i + 1)$. Typically, $M$ is one to two orders of magnitude smaller than $D_i$ and $D_a$, which are commonly equal to each other, making the proposed approach more efficient in parameters and operations.

There is a connection between EP and *slot attention* (Locatello et al., 2020), where slots are also multiple vectors in the input feature space. To derive EP from slot attention, one needs to perform only a single iteration; remove LayerNorm, GRU and MLP; make slot vectors learnable rather than initialized at random; and concatenate the output features into a global representation of appropriate dimension. Thus, EP can be seen as a lightweight counterpart of slot attention, where the absence of interactions is compensated by the query features being learned.

### 3.3 EXISTING VARIANTS

We analyze existing methods as instances of the presented framework (Table 1), and examine common variants, considering their relationship to the framework despite slight deviations. Additional methods considered in experiments are presented in Appendix B.

**AbMILP** (Rymarczyk et al., 2021) is the simplest variant. It fixes $W_K$ and $W_V$ to identity and is equivalent to MHCA with a learnable query feature framework in (7), with a single head ($M = 1$). It can also be seen as a special case of our proposed method in (10) with one learnable query feature, i.e. $M = 1$. AbMILP requires only $D_i$ parameters and computes attention with a single matrix-vector multiplication, but its performance is limited by the single head/query.

**AIM** (El-Nouby et al., 2024) is an instance of MHCA with a learnable query feature. It deviates from the generic framework by applying batch normalization on the input features. It does follow (2) and (7) with $M$ heads and $D_a = D_i = D_o$, but replaces $X$ by $\mathrm{BN}(X)$. Batch normalization introduces minor additional parameters and a slight computational overhead compared to the default variant of the framework.

**DELF** (Noh et al., 2017) feeds each of the $N$ input features to a MLP whose output is a scalar attention value in $[0, 1]$. It can be viewed as an instance of the MHCA with a learnable query feature and $M = 1$ with the following modifications. Key and value projection matrices share the same weights, i.e. $W_K = W_V = W$, a non-linearity is introduced in equation (7) by $\hat{\boldsymbol{a}} = \mathrm{ReLU}(WX)^\top \boldsymbol{q}$, where subscript $j$ is skipped due to $M = 1$, and softmax in (5) is replaced by element-wise softplus, $\boldsymbol{a} = \sigma_p(\hat{\boldsymbol{a}})$. In the context of DELF, the query feature $\boldsymbol{q}$ can be seen as the parameter of a $1 \times 1$ convolutional layer. DOLG (Yang et al., 2021) adopts a similar design choice.

**SimPool** (Psomas et al., 2023) can be seen as an instance of MHCA with a single head ($M = 1$) that uses a data-dependent input vector $\boldsymbol{u}$, $W_V$ fixed to identity, and layer normalization on the input features. Specifically, the query feature is obtained as $\boldsymbol{q} = W_Q \boldsymbol{u}$, $\boldsymbol{u} = \frac{1}{N} X^\top \mathbf{1}$ ($M = 1$). $X$ is replaced by $\mathrm{LN}(X)$ for key and value transforms. Compared to MHCA with a learnable query, SimPool saves $D_i$ parameters and has the same complexity.

**V-JEPA.** The first part of V-JEPA (Bardes et al., 2024) is identical to the MHCA framework but applies layer normalization on the input features for key and value transforms, like SimPool. Its second part is an MLP with GeLU activation (Hendrycks & Gimpel, 2016) and residual connections, making the overall process equivalent to a transformer block.

Table 1: *Attentive pooling variants as aligned algorithmic steps,* fitting the framework of Sec. 3.1–3.2. We list (i) how the query is formed, (ii) key/value transforms, (iii) attention predictor outputs, and (iv) pooling operation. $\sigma_m$: softmax, $\sigma_p$: softplus. $\phi(x) := W_2 \,\mathrm{GeLU}(W_1 x)$. blue: learnable.

| METHOD | QUERY SOURCE | KEY TRANSFORM | VALUE TRANSFORM | ATTENTION | POOLING |
|---|---|---|---|---|---|
| MHCA | $\boldsymbol{q}_j \in \mathbb{R}^{d_a}$ | $K_j = W_{K_j} X$ | $V_j = W_{V_j} X$ | $\boldsymbol{a}_j = \sigma_m(K_j^\top \boldsymbol{q}_j)$ | $\boldsymbol{y}_j = V_j \boldsymbol{a}_j$ |
| AbMILP | $\boldsymbol{q} \in \mathbb{R}^{D_i}$ | $K = X$ | $V = X$ | $\boldsymbol{a} = \sigma_m(K^\top \boldsymbol{q})$ | $\boldsymbol{y} = V \boldsymbol{a}$ |
| AIM | $\boldsymbol{q}_j \in \mathbb{R}^{d_a}$ | $K_j = W_{K_j} \mathrm{BN}(X)$ | $V_j = W_{V_j} \mathrm{BN}(X)$ | $\boldsymbol{a}_j = \sigma_m(K_j^\top \boldsymbol{q}_j)$ | $\boldsymbol{y}_j = V_j \boldsymbol{a}_j$ |
| DELF | $\boldsymbol{q} \in \mathbb{R}^{D_i}$ | $K = \mathrm{ReLU}(WX)$ | $V = WX$ | $\boldsymbol{a} = \sigma_m(K^\top \boldsymbol{q})$ | $\boldsymbol{y} = V \boldsymbol{a}$ |
| SimPool | $\boldsymbol{q} = W_Q \boldsymbol{u} \in \mathbb{R}^{D_i}, \boldsymbol{u} = \frac{1}{N} X^\top \mathbf{1}$ | $K = W_K \mathrm{LN}(X)$ | $V = \mathrm{LN}(X)$ | $\boldsymbol{a} = \sigma_m(K^\top \boldsymbol{q})$ | $\boldsymbol{y} = V \boldsymbol{a}$ |
| V-JEPA | $\boldsymbol{q}_j = W_{Q_j} \boldsymbol{u} \in \mathbb{R}^{d_a}, \boldsymbol{u} \in \mathbb{R}^{D_i}$ | $K_j = W_{K_j} \mathrm{LN}(X)$ | $V_j = W_{V_j} \mathrm{LN}(X)$ | $\boldsymbol{a}_j = \sigma_m(K_j^\top \boldsymbol{q}_j)$ | $\boldsymbol{y} = \phi(W_P V_j \boldsymbol{a}_j)$ |
| EP (ours) | $\boldsymbol{q}_j \in \mathbb{R}^{D_i}$ | $K = X$ | $V_j = W_{V_j} X$ | $\boldsymbol{a}_j = \sigma_m(K^\top \boldsymbol{q}_j)$ | $\boldsymbol{y}_j = V_j \boldsymbol{a}_j$ |

## 4 EXPERIMENTS

### 4.1 EXPERIMENTAL SETUP

**Datasets.** We evaluate attentive probing across diverse image classification benchmarks, including ImageNet-1K (IN-1K) (Deng et al., 2009), CIFAR-100 (Krizhevsky et al., 2009), Places365 (Zhou et al., 2014), CUB-200 (Wah et al., 2011), FGVC Aircraft (Maji et al., 2013), Stanford Cars (Krause et al., 2013), and Food-101 (Bossard et al., 2014). More details in subsection C.1.

**Pre-training methods.** We conduct attentive probing experiments with frozen models from diverse pre-training paradigms: MIM (e.g., MAE and CAPI), JEA (e.g., BYOL and DINO), hybrid (e.g., iBOT and DINOv2), VLMs (e.g., CLIP and SigLIP), and generative (e.g., DiT and AIMv2). Model sizes vary from small (e.g. ViT-S for MAE) to extra large (e.g. DiT-XL for DiT).

**Pooling/probing methods.** We compare attentive probing against a diverse set of methods, covering different paradigms. First, we evaluate attentive poolings originally designed for probing, including AIM (El-Nouby et al., 2024), CAE (Chen et al., 2023), CAPI (Darcet et al., 2025), and V-JEPA (Bardes et al., 2024). Second, we include attentive poolings originally proposed in other contexts but applicable to probing, such as AbMILP (Rymarczyk et al., 2021), SimPool (Psomas et al., 2023), CLIP (Radford et al., 2021), SigLIP (Zhai et al., 2023; Wightman, 2019), CoCa (Yu et al., 2022), CaiT (Touvron et al., 2021), and DELF (Noh et al., 2017). Additionally, we include feature re-weighting methods like CBAM (Woo et al., 2018), applying global average pooling to obtain the global descriptor. As baselines, we use `[CLS]`, which corresponds to standard linear probing when the backbone provides a classification token, and GAP, which serves as linear probing when no `[CLS]` token is available, but can also be interpreted as a baseline attentive probing method with uniform attention over patch tokens. To establish a reference, we also evaluate a ViT (Dosovitskiy et al., 2021) block, applying global average pooling to extract the global representation, and MHCA. All methods operate on the same input features—namely, the patch tokens extracted from the frozen backbone—ensuring a fair and consistent comparison. Unless otherwise stated, $D_o=D_i=D_a$.

**Parameter-efficient fine-tuning (PEFT) methods.** We compare attentive probing against PEFT methods on ImageNet-1K: LoRA (Hu et al., 2022), BitFit (Ben Zaken et al., 2022), and LayerNorm tuning (Zhao et al.).

**Evaluation protocols.** Attentive probing is performed for 90 epochs. We evaluate top-1 classification accuracy on the validation set of each dataset. Additionally, we compute the number of parameters and measure the FLOPs to assess computational efficiency and scalability.

### 4.2 EXPERIMENTAL RESULTS

**Accuracy vs. parameters.** In Figure 2, we compare efficient probing (EP) with baseline and competitor methods using MAE ViT-B, BEiTv2 ViT-B, and CAPI ViT-L on ImageNet-1K, and MAE ViT-B on Food-101. We plot top-1 accuracy against the number of trainable parameters, including both attentive pooling and classifier parameters, and overlay the Pareto frontier to highlight optimal trade-offs. The two primary baselines, `[CLS]` and GAP, are the most parameter-efficient, as they introduce no overhead beyond the classifier parameters, but yield noticeably lower accuracy. In contrast, methods like V-JEPA, CaiT, SigLIP, and the reference ViT block employ significantly more parameters, though within the attentive probing setting, their increased complexity provides mostly marginal accuracy improvements. Among the existing attentive probing or pooling methods, SimPool provides moderate accuracy but is not particularly parameter-efficient, while CAE and CLIP achieve stronger performance at the cost of higher parameter counts. AbMILP, DELF, AIM, and MHCA lie primarily on the Pareto frontier, striking the optimal balance.

EP consistently achieves the best accuracy–parameter trade-off, positioning itself on the left or upper-left region of the Pareto frontier across pre-training methods and datasets. A key factor is its flexibility in controlling the number of queries $M$ and the output dimensionality $D_o$ (because of $W_V$), allowing adaptation to different parameter constraints. Notably, on ImageNet-1K with MAE ViT-B, $EP_{64}$ (64 queries) achieves a state-of-the-art top-1 accuracy of 75.6% with less than 1.4M parameters. $EP_{48}$ with 48 queries and $D_o = D_i/8$ achieves 70.3% top-1 accuracy, while having a little more than 200k parameters, i.e. almost 4× less than linear probing (`[CLS]`). On ImageNet-1K with BEiTv2 ViT-B and with CAPI ViT-L, $EP_{32}$ and $EP_{64}$ achieve a state-of-the-art accuracy of

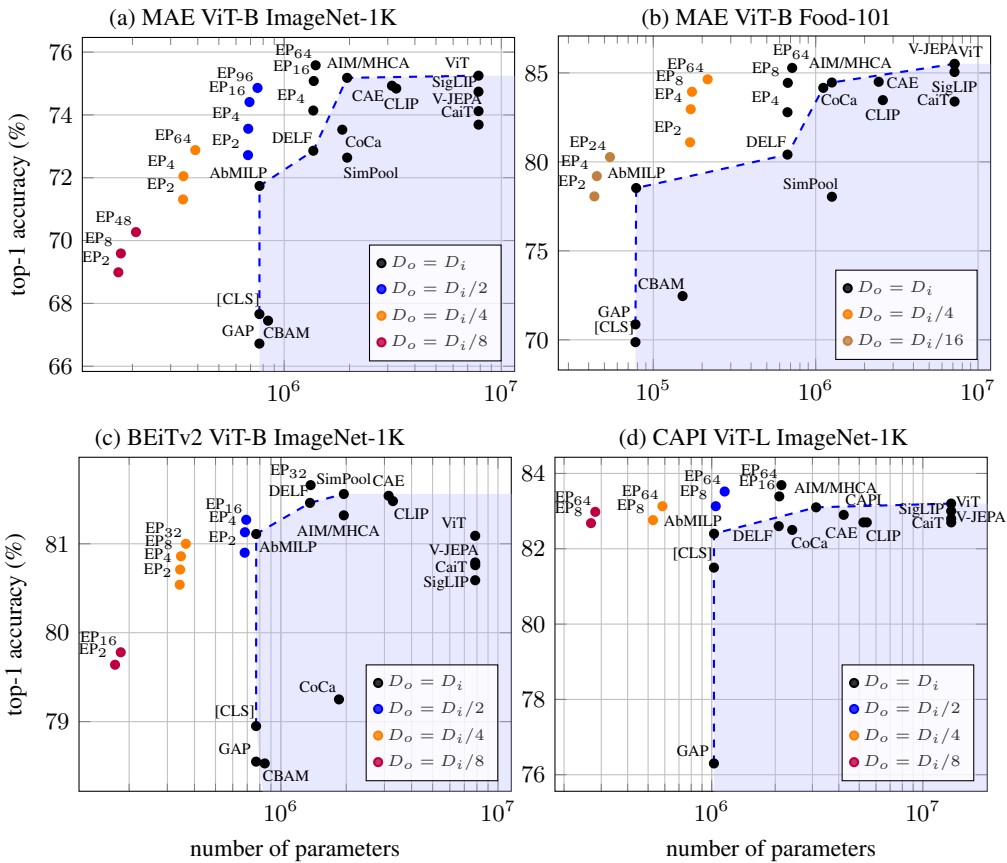

Figure 2: *Top-1 classification accuracy vs. number of parameters* for various self-supervised pre-training methods across different datasets. We evaluate both dedicated probing mechanisms (e.g., V-JEPA) and repurposed attentive pooling methods (e.g., CLIP). EP variants are marked with different colors for different output dimensionalities $D_o$. $\text{EP}_M$: efficient probing with $M$ learnable queries. [CLS]: linear probing using the classification token; GAP: global average pooling over patch tokens; MHCA: multi-head cross-attention; ViT: default transformer block.

81.7% and 83.7% respectively. On Food-101, a dataset smaller than ImageNet-1K, we observe that reducing $D_o$ even to $D_i/16$ does not significantly hurt performance. More results in subsection C.2.

**Accuracy vs. computational cost.** In Figure 3, we compare different pooling/probing methods in terms of top-1 accuracy against computational cost, measured in GFLOPs. Baselines [CLS] and GAP are the most efficient but less accurate, while self-attention methods (e.g., ViT, CLIP) incur higher cost due to additional attention computations. EP achieves better accuracy than a ViT block with over $10\times$ less compute, and consistently lies on the Pareto frontier, combining high accuracy with low cost. By adjusting the output dimensionality $D_o$, EP scales to different computational budgets without significant accuracy loss. Note that backbone FLOPs (e.g., 17.58 GFLOPs for MAE ViT-B) are not shown, since they are identical across all methods and thus omitted for fairness. Here we only compare the additional cost of the pooling/probing methods.

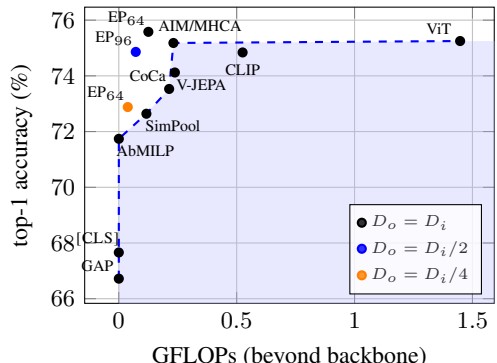

Figure 3: *Top-1 classification accuracy vs. GFLOPs* for MAE ViT-B with different probings on ImageNet-1K.

Table 2: *Comparison of pre-training methods in terms of different evaluation protocols* on ImageNet-1K. # PAR.: number of trainable parameters including classifier; †Results reported from the official paper under different than probing setup (e.g., augmentations); ‡Evaluated using GAP.

| | METHOD | ARCH | PRE-TRAINING | k-NN | LP | # PAR. | EP | # PAR. | GAIN | FT† | # PAR. |
|---|---|---|---|---|---|---|---|---|---|---|---|
| MIM | MAE (He et al., 2022) | ViT-S/16 | IN-1K | 26.7 | 47.4 | 0.4M | 64.6 | 0.5M | +17.2 | 80.6 | 22M |
| | | ViT-B/16 | | 46.1 | 67.7 | 0.8M | 75.6 | 1.4M | +7.9 | 83.6 | 87M |
| | | ViT-L/16 | | 58.2 | 76.0 | 1.0M | 79.3 | 2.1M | +3.3 | 85.9 | 304M |
| | BEiTv2 (Peng et al., 2022) | ViT-B/16 | IN-1K | 74.8 | 79.0 | 0.8M | 81.7 | 1.4M | +2.7 | 85.0 | 87M |
| | SimMIM (Xie et al., 2022) | ViT-B/16 | IN-1K | 15.1 | 51.5 | 0.8M | 65.1 | 1.4M | +13.6 | 83.8 | 87M |
| | CAPI (Darcet et al., 2025) | ViT-L/14 | IN-1K | 76.7 | 81.5 | 1.0M | 83.6 | 2.1M | +2.1 | × | 304M |
| JEA | BYOL (Grill et al., 2020) | RN-50 | IN-1K | 64.8 | 74.3 | 2.0M | 75.1 | 6.3M | +0.8 | 77.7 | 26M |
| | DINO (Caron et al., 2021) | ViT-B/16 | IN-1K | 76.1 | 77.3 | 0.8M | 77.8 | 1.4M | +0.5 | 82.8 | 87M |
| HYBRID | iBOT (Zhou et al., 2022) | ViT-B/16 | IN-1K | 77.0 | 78.7 | 0.8M | 79.2 | 1.4M | +0.5 | 84.0 | 87M |
| | DINOv2 (Oquab et al., 2024) | ViT-B/14 | LVD-142M | 81.8 | 83.2 | 0.8M | 84.0 | 1.4M | +0.8 | × | 87M |
| | | ViT-L/14 | | 83.5 | 85.2 | 1.0M | 85.6 | 2.1M | +0.4 | × | 304M |
| | Franca (Venkataramanan et al., 2025) | ViT-L/14 | IN-21k | 82.2 | 83.8 | 1.0M | 84.3 | 2.1M | +0.5 | × | 304M |
| | DINOv3 (Siméoni et al., 2025) | ViT-B/16 | LVD-1689M | 83.0 | 84.0 | 0.8M | 84.4 | 1.4M | +0.4 | × | 87M |
| | | ViT-L/16 | | 85.3 | 86.6 | 1.0M | 87.1 | 2.1M | +0.5 | × | 304M |
| VLM | CLIP (Radford et al., 2021) | ViT-L/14 | WIT | 77.2 | 82.3 | 0.8M | 83.4 | 2.1M | +1.1 | × | 305M |
| | SigLIP (Zhai et al., 2023) | ViT-L/16 | WebLI | 83.7 | 84.1‡ | 1.0M | 86.1 | 2.1M | +2.0 | × | 305M |
| | SigLIP2 (Tschannen et al., 2025) | ViT-L/16 | WebLI | 84.4 | 85.2‡ | 1.0M | 87.0 | 2.1M | +1.8 | × | 305M |
| GEN | DiT (Peebles & Xie, 2023) | DiT-XL/2 | IN-1K | 8.3 | 32.7‡ | 1.2M | 57.0 | 2.5M | +24.3 | × | 676M |
| | AIMv2 (Fini et al., 2025) | ViT-L/14 | custom* | 80.8 | 84.8‡ | 1.0M | 85.9 | 2.1M | +1.1 | × | 304M |

*Note.* custom*: DFN-2B (Fang et al., 2024), COYO (Byeon et al., 2022), HQITP (Fini et al., 2025). Default EP = EP$_{32}$.

Figure 4: *Accuracy–parameter trade-off of probing and parameter-efficient fine-tuning methods* on ImageNet-1K using a frozen MAE ViT-B/16 backbone. The *hybrid* LoRA + EP configurations achieve a *strictly better trade-off* than both pure EP and pure LoRA.

**Comparison of pre-training methods.** Table 2 compares methods from diverse pre-training paradigms under multiple evaluation protocols on ImageNet-1K. Fine-tuning (FT) achieves the highest accuracy but is compute-intensive and unsustainable at scale, making it increasingly rare in recent evaluations (×). $k$-NN consistently underperforms, reflecting the weak separability of raw features. Linear probing (LP) offers a stronger baseline, yet EP provides consistent gains across all paradigms, often with only a small increase in parameters. Notably, EP changes the relative ranking of methods compared to LP and $k$-NN: for example, MAE surpasses BYOL, and CAPI outperforms CLIP, challenging the view that MIM methods are weaker. In general, methods whose pre-training optimizes representations of patch tokens rather than an explicit global representation are benefit the most from attentive probing (e.g., SimMIM +13.6%, DiT +24.3%). We further analyze this trend in subsection C.3. Across the board, EP narrows the gap to full fine-tuning while remaining lightweight and scalable, demonstrating both its generality and effectiveness.

**Comparison against PEFT methods.** In Figure 4 we compare the most efficient baseline probing methods (`[CLS]`, AbMILP, DELF, AIM), EP at different output dimensionalities $D_o$, more than 40 LoRA variants, BitFit, and LayerNorm tuning, all on a frozen MAE ViT-B/16. We sweep LoRA across individual transformer layers (4, 8, 12) as well as all layers, and across multiple configurations (e.g. $W_Q$-only, $W_K$-only, $W_V$-only, $W_Q$+$W_K$, $W_Q$+$W_K$+$W_V$) with ranks $\rho \in \{8, 16, 32, 64\}$.

Overall, LoRA applied to a small subset of layers (red, cyan, magenta crosses) lies roughly on or slightly above the baseline Pareto front, but is consistently dominated by EP in the accu-

racy–parameter plane. For instance, $EP^{D_i/2}$ attains 74.9% top-1 accuracy with only 750K parameters, whereas the best single-layer LoRA configuration around that region requires around 1.2M parameters. LayerNorm tuning and BitFit (green and brown crosses) also outperform the pure probing baselines, yet EP remains strictly more efficient. All-layer LoRA configurations (orange crosses) move closer to the EP Pareto curve and some even surpass it in accuracy (up to 76.7%). In subsection C.2, we present an analysis with in- and out-of-domain $k$-NN experiments on frozen features, showing that all-layer LoRA modifies the representation more strongly than EP, consistent with their roles as task-adaptive low-rank fine-tuning (LoRA) versus representation-preserving probing (EP).

Motivated by these observations, we finally combine one of the most parameter-efficient LoRA variants (LoRA on all $W_V$ matrices across layers) with EP for different output dimensionalities $D_o$ (orange star–cross markers). The resulting LoRA+EP configurations form a new dominant region in the accuracy–parameter plane, *strictly improving over both pure EP and pure LoRA*. For example, a hybrid setting with 850K parameters achieves 76.99% top-1 accuracy, improving over both the best pure EP configuration (75.58% with 1.38M parameters) and the best all-layer LoRA variant (76.72% with 1.95M parameters). At the low-parameter end, a LoRA+EP configuration with only 250K parameters already reaches 71.99% accuracy, i.e. about 4.3% above [CLS] linear probing (67.66%) while using over 3× fewer parameters. These results indicate that *EP captures information that LoRA alone does not*, and vice versa: rather than being redundant with PEFT, EP provides a complementary, parameter-efficient probing (PEP) mechanism that remains beneficial.

## 4.3 EXPERIMENTAL ANALYSIS

**Impact of $W_K$ and $W_V$.** Our analysis in subsection 3.2 posits that while a single learnable query $q$ effectively absorbs the key transformation $W_K$ in single-head attention, the same does not hold in multi-head. To empirically validate this, we probe MAE ViT-B pre-trained on ImageNet-1K with four variants: single-head ($M{=}1$) vs. multi-head ($M{>}1$), each with (7) vs. without $W_K$ (8). Specifically, we evaluate AbMILP and AIM for the single-head and multi-head (AIM$_{12}$ with 12 heads) case, respectively. In single-head attention, removing $W_K$ has minimal impact on performance (71.8% → 71.7%), while in multi-head the drop is noticeable (75.1% → 72.9%), since each query now interacts only with a subspace. Note that MHCA (7), that natively includes $W_K$, in AIM$_{12}$ and our MQCA (10), that natively does not includes $W_K$, in EP$_{12}$, configured with the same number of heads/queries ($M{=}12$), attain identical accuracy (75.1%), reflecting their mathematical equivalence. However, EP$_{12}$ reaches this performance with far fewer parameters (1.36M for EP$_{12}$ vs. 1.95M for AIM$_{12}$), making its design more parameter-efficient. We also ablate the effect of the value transformation $W_V$, which operates on patch tokens (2), by adding or removing it across pooling methods. Introducing $W_V$ to GAP results in a top-1 accuracy improvement from 66.7% → 68.0%. Conversely, removing $W_V$ from EP$_{12}$ degrades performance from 75.1% → 72.1%. A similar accuracy drop is observed for other methods, such as AIM (75.1% → 72.0%) and CAE (74.9% → 72.2%), confirming that $W_V$ is a critical component.

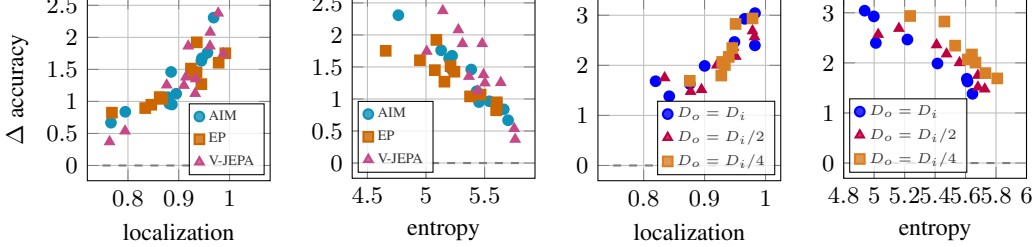

Figure 5: *Classification accuracy vs. attention quality* on ImageNet-1K. Each point corresponds to an attention predictor (head or query). $\Delta$ accuracy measures the drop when replacing an attention predictor's distribution with uniform. Plots show relations to localization quality (1st, 3rd) and entropy (2nd, 4th). Left: different attentive probing methods; Right: varying $D_o$ for EP.

**Classification vs. localization.** We investigate whether the quality of attention maps in terms of localization contributes positively to classification accuracy. Localization quality is measured by (i) the attention mass within the ground-truth bounding box (Deng et al., 2009) and (ii) the entropy of the attention distribution, averaged over the validation set. To estimate each predictor's contribution

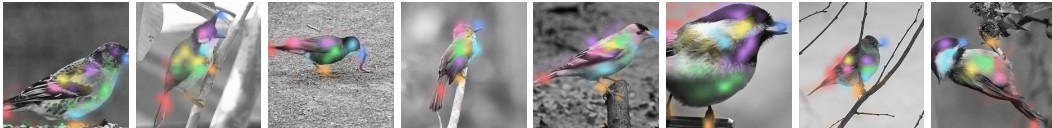

Figure 6: *Attention maps of* $EP_8$. Each query is assigned a distinct color. Semantic correspondences emerge (e.g., tails, beaks, feet), with each query capturing complementary regions and enabling a structured decomposition of visual cues. MAE ViT-B pre-trained on IN-1K, probed with EP.

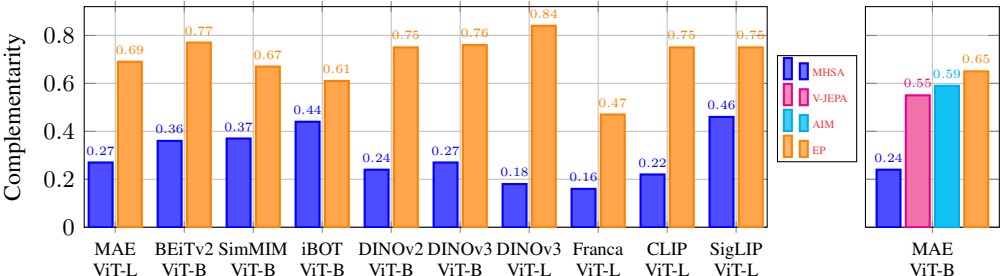

Figure 7: *Complementarity scores of attention maps across different backbones and probings*. We compare the diversity of the internal MHSA heads in the last block against the external EP queries (left), V-JEPA heads, and AIM heads (right). Number of predictors are matched for fairness.

to classification, we measure the accuracy drop when replacing its attention with a uniform distribution. Figure 5 shows a clear correlation: predictors with better localization and lower entropy exert stronger influence on accuracy. This holds across different attentive methods as well as EP with reduced output dimensionality $D_o$. As also seen in Figure 2, lowering $D_o$ degrades performance; the rightmost plots further suggest that this effect arises not only from reduced representational capacity but also from diminished attention quality, as reflected in higher entropy.

**Attention complementarity.** In Figure 6, we jointly visualize the attention maps of $EP_8$. An emerging property of EP is that its queries specialize in different object regions, yielding complementary and interpretable attention patterns. Queries consistently attend to distinct parts, producing semantic correspondences across images and a structured decomposition of visual cues. To quantify this diversity, we define a *complementarity* metric measuring how differently attention predictors distribute mass over patch tokens. For standard backbones, we extract the [CLS]→patch attention from the internal last-block MHSA heads, and for EP the learned queries, with matched counts for fairness. We $L2$-normalize each distribution, compute pairwise cosine similarities, and define complementarity as one minus the average off-diagonal similarity. Higher values indicate more diverse (less redundant) attention. As shown in Figure 7, EP achieves significantly higher complementarity than MHSA and outperforms other probing approaches. More experiments in subsection C.3.

## 5 CONCLUSION

We revisit evaluation protocols for pre-training methods and introduce efficient probing (EP), a scalable alternative to fine-tuning. Lightweight yet expressive, EP delivers interpretable attention, strong generalization across paradigms, and consistent gains over linear probing—up to +24.3% on ImageNet-1K. By comparing EP to standard parameter-efficient fine-tuning (PEFT) baselines, we show that EP is not only competitive on its own but also complementary: hybrid configurations achieve the best accuracy–parameter trade-offs, hinting at new research directions.

**Acknowledgments.** This work was supported by the EU Horizon Europe programme MSCA PF RAVIOLI (No. 101205297) and the Junior Star GACR GM 21-28830M. We acknowledge VSB–Technical University of Ostrava, IT4Innovations National Supercomputing Center, Czech Republic, for awarding this project (OPEN-33-67) access to the LUMI supercomputer, owned by the EuroHPC Joint Undertaking, hosted by CSC (Finland) and the LUMI consortium, through the Ministry of Education, Youth and Sports of the Czech Republic via the e-INFRA CZ project (ID: 90254). AWS resources were provided by the National Infrastructures for Research and Technology GRNET and funded by the EU Recovery and Resiliency Facility.

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

APPENDIX

# A    ADDITIONAL RELATED WORKS

## A.1    POOLING

Pooling reduces spatial resolution while retaining semantic information. In CNNs, fixed pooling (e.g., global average pooling (Lin et al., 2013; He et al., 2016)) is standard; in vision transformers (ViTs) (Dosovitskiy et al., 2021), the `[CLS]` token aggregates features via self-attention.

Recent work proposes attention-based pooling to enhance representation quality. SimPool (Psomas et al., 2023) replaces global average pooling using trainable attention in both CNNs and ViTs. Vision-language models such as CLIP (Radford et al., 2021), SigLIP (Wightman, 2019), and CoCa (Yu et al., 2022) use attentive pooling or cross-attention to fuse modalities. V-JEPA (Bardes et al., 2024) applies cross-attention pooling for probing pretrained representations. In image retrieval, DELF (Noh et al., 2017) and DOLG (Yang et al., 2021) use spatial attention to focus on salient regions. CaiT (Touvron et al., 2021) improves class-token attention, AbMILP (Rymarczyk et al., 2021) uses single-query pooling for multiple-instance learning, and CBAM (Woo et al., 2018) combines channel and spatial attention to recalibrate features. Although these poolings are originally introduced in diverse contexts, we repurpose them for probing of frozen pre-trained models, enabling a fair and comprehensive benchmark.

## A.2    PARAMETER-EFFICIENT FINE-TUNING

Parameter-efficient fine-tuning (PEFT) adapts large pre-trained models to downstream tasks without updating all model parameters. PEFT techniques broadly include *additive*, *selective*, and *low-rank adaptation* methods.

**Additive.**    Additive methods introduce small, task-specific modules into the frozen backbone, leaving the pre-trained weights untouched. These modules often reside within the transformer blocks and are trained to specialize the model for a new domain. Notable examples include Adapter-Fusion (Pfeiffer et al., 2021), LeTS (Fu et al., 2021), and TADA (Hung et al., 2023) in natural language processing (NLP), VPT (Jia et al., 2022), AdaptFormer (Chen et al., 2022), and Adapter-X (Li et al., 2024) in computer vision (CV), and FMA (He et al., 2024), AiRs (Hu et al., 2024), and DEFLECT (Thoreau et al., 2025) in remote sensing (RS).

**Selective.**    Selective methods fine-tune only specific subsets of parameters, typically chosen based on their functional role or estimated importance. Examples include BitFit (Ben Zaken et al., 2022), which adjusts only the bias terms, and norm-tuning approaches (Zhao et al.) which update only the normalization layers. These techniques avoid introducing new components, making them lightweight, though sometimes at the expense of performance.

**Low-rank adaptation.**    Low-rank adaptation methods like LoRA (Hu et al., 2022) in NLP assume that parameter updates lie in a low-dimensional subspace. They inject trainable low-rank matrices into existing layers, yielding strong performance with minimal parameter growth. In the vision domain, LoRa and its variants have been effectively adapted to vision transformers (ViTs), often rethinking where and how low-rank modules are inserted to align with the spatial and hierarchical nature of visual representations. Notable examples include structure-aware methods like Serial LoRA (Zhong et al., 2025) and Flat-LoRA (Li et al., 2025), layer-wise extensions such as AdaptFormer (Chen et al., 2022), and task specific designs like PETAH (Augustin et al., 2025) and MeLo (Zhu et al., 2024), which adapt LoRa to mobile inference and medical imaging, respectively. Continued pretraining approaches such as ExPLoRA (Khanna et al., 2025) further extend low-rank adaptation to domain-shifted self-supervised settings.

EP naturally fits the additive PEFT family. It introduces a compact learnable query set interacting with frozen tokens via multi-head cross-attention. Unlike typical prompt-based methods, it avoids backbone modifications and focuses training on minimal parameters. Thus, EP efficiently combines additive PEFT simplicity with task-specific attentive pooling.

## B  ADDITIONAL METHODS

We provide here additional attentive pooling/probing methods evaluated in subsection 4.2 but not described in detail in subsection 3.3. These approaches represent variations of the MHCA framework, highlighting only their key deviations from the default design.

**CLIP.**  CLIP (Radford et al., 2021) differs from MHCA by employing self-attention rather than cross-attention. Specifically, CLIP prepends a global average pooled (GAP) token to the layer-normalized input features, treating this token as a global representation. All tokens, including the GAP token, are augmented with learnable positional encodings and processed through a single self-attention block (which includes a query projection matrix $W_Q$). The global representation is extracted from the output corresponding to the GAP token. Additionally, CLIP includes a linear projection matrix $W_P$ after attention aggregation. These modifications enable interactions across all tokens but increase parameter count and computational complexity.

**CAiT.**  CAiT (Touvron et al., 2021) adapts the MHCA-with-learnable-query formulation by concatenating the learnable query token with the input features and applying self-attention rather than cross-attention. It retains the query projection matrix $W_Q$ and includes a linear projection matrix $W_P$ after attention aggregation, followed by an MLP block with GELU activations, residual connections, and LayerScale parameters, similar to V-JEPA. The global representation is obtained from the updated query token after these operations, thus increasing complexity and parameter count relative to the default MHCA variant.

**SigLIP.**  SigLIP (Zhai et al., 2023; Wightman, 2019) remains close to the MHCA-with-learnable-query formulation but retains the query projection matrix $W_Q$. After the attention aggregation, SigLIP incorporates an output projection $W_P$, followed by a transformer-style MLP block with GELU activation and residual connections, similar to V-JEPA and CAiT. Optional layer normalization can also be applied before the MLP. These changes add further parameters and computational overhead compared to the baseline MHCA design.

**CAE.**  CAE (Chen et al., 2023) follows the MHCA-with-learnable-query template closely but retains the query projection matrix $W_Q$ and applies separate layer normalization to both input features and the query token prior to attention. After attention aggregation, it employs an additional output projection matrix $W_P$. These modifications introduce additional parameters and computational complexity.

**CoCa**  CoCa (Yu et al., 2022) is aligned with the MHCA-with-learnable-query framework but retains the query projection matrix $W_Q$ and layer-normalizes the query token before computing attention. Attention and value aggregation both occur in a reduced-dimensional space, with dimension $D_a = D_o < D_i$. A final linear projection matrix $W_{\text{proj}}$ is then applied to restore the feature dimension to the original backbone dimension $D_i$. These choices introduce a controlled amount of additional complexity and parameters.

In Figure 8 we present a visual comparison of three selected attentive pooling/probing techniques: AbMILP (Rymarczyk et al., 2021), AIM (El-Nouby et al., 2024), and V-JEPA (Bardes et al., 2024). AbMILP (top-left) serves as a lightweight method, employing a single-head learnable query without additional linear projection matrices, thus requiring only $D_i$ parameters. AIM (top-right) extends this by adopting multi-head cross-attention, operating within multiple subspaces. This approach introduces linear projection matrices for keys and values, increasing the number of parameters, yet allowing more expressive query-key interactions. V-JEPA (bottom) represents a significantly more complex and computationally intensive architecture. Beyond multi-head attention and multiple linear projections for queries, keys, and values, it integrates an additional projection step, followed by a multi-layer perceptron (MLP) featuring GeLU activation and residual connections.

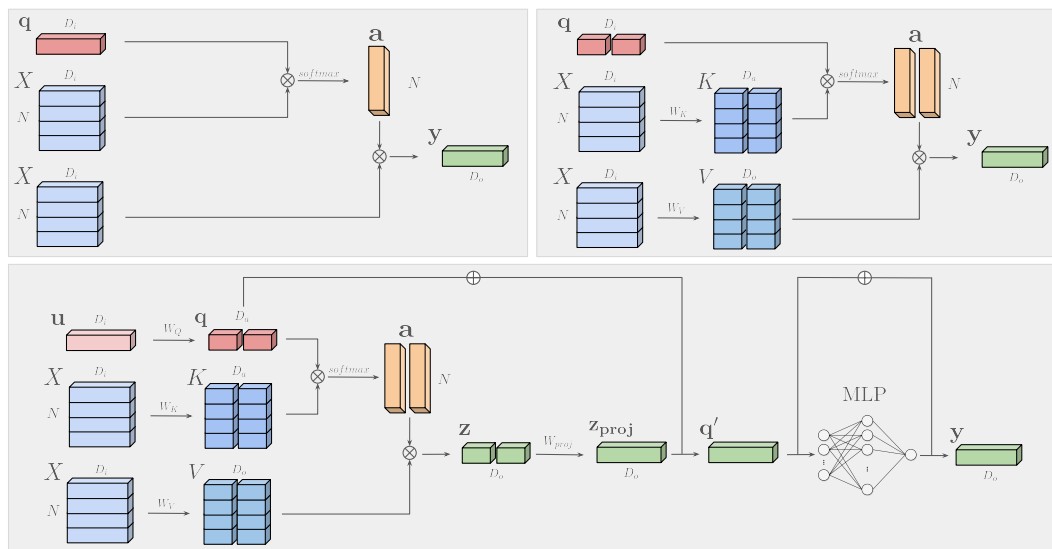

Figure 8: *Visual comparison of three attentive pooling/probing methods*. AbMILP (top-left) employs a single-head, learnable query without linear projections, minimizing complexity. AIM (top-right) extends the approach by introducing multi-head attention, operating in multiple subspaces, and applies linear projections to keys and values. V-JEPA (bottom) offers a more comprehensive architecture by integrating multi-head attention with extensive linear projections and an additional MLP block with a residual connection, increasing representational capacity.

## C ADDITIONAL EXPERIMENTS

### C.1 EXPERIMENTAL SETUP

**Datasets.** We evaluate attentive probing across diverse image classification benchmarks. As a large-scale dataset, ImageNet-1K (Deng et al., 2009) serves as the primary benchmark, containing 1.28M images across 1,000 categories. CIFAR-100 (Krizhevsky et al., 2009) provides a smaller yet challenging 100-class task with 60K images. To assess scene understanding, we use Places365 (Zhou et al., 2014), comprising 1.8M images spanning 365 scene types. For fine-grained classification, we evaluate on CUB-200 (Wah et al., 2011) (11,788 images, 200 bird species), FGVC Aircraft (Maji et al., 2013) (10K images, 100 aircraft models), Stanford Cars (Krause et al., 2013) (16K images, 196 car types), and Food-101 (Bossard et al., 2014) (101K images, 101 food categories). Together, these datasets span a wide spectrum of scales and challenges—large-scale vs. small-scale, generic vs. fine-grained, and object- vs. scene-centric—providing a comprehensive testbed for probing methods.

**Pooling/probing methods.** We adopt AbMILP with `depth=1`, which reduces AbMILP to the formulation described in Table 1 and subsection 3.3. In Table 3, we evaluate AbMILP on ImageNet-1K with larger depths (`depth=2,3`), which correspond to the MLP-based variants and scale as $\mathcal{O}(D_i^2)$, exploring their parameter–accuracy tradeoffs. Although deeper MLPs

Table 3: *AbMILP architecture ablation* increasing MLP depth. # PAR.: number of parameters.

| METHOD | # PAR. | ACCURACY |
|---|---|---|
| AbMILP (depth 1) | 769,769 | 71.74 |
| AbMILP (depth 2) | 1,360,361 | 72.25 |
| AbMILP (depth 3) | 1,950,953 | 72.84 |

introduce substantially more parameters ($+\mathcal{O}(D_i^2)$), the gains in accuracy are marginal. To ensure a fair comparison in the *accuracy-vs.-parameter-efficiency* setting, we consider the most competitive AbMILP variant (`depth=1`) as default, which successfully lies on the Pareto front.

**Implementation details.** We evaluate 15 models spanning five pre-training paradigms: four masked image modeling (MAE (He et al., 2022), BEiTv2 (Peng et al., 2022), SimMIM (Xie et al., 2022), CAPI (Darcet et al., 2025)), two joint-embedding (BYOL (Grill et al., 2020), DINO (Caron et al., 2021)), two hybrid (iBOT (Zhou et al., 2022), DINOv2 (Oquab et al., 2024)), three vision-language (CLIP (Radford et al., 2021), SigLIP (Zhai et al., 2023), SigLIP2 (Tschannen et al., 2025)),

and two generative—DiT (diffusion) (Peebles & Xie, 2023) and AIMv2 (autoregressive) (Fini et al., 2025). Architectures range from small (e.g., ViT-S for MAE) to extra-large (e.g., DiT-XL for DiT).

To ensure a fair comparison of pooling/probing methods, we mostly adopt the LARS optimizer You et al. (2017) and conduct a learning rate search in the range [0.1, 5.0] with a step size of 0.1 for each model. For large-scale datasets such as ImageNet-1K and Places365, we fix the learning rate to 0.1 due to the computational cost of an exhaustive search. All models are trained for 90 epochs with 10 warm-up epochs, ensuring consistent training schedules even though many models converge much earlier (e.g., SigLIP within 15 epochs). The effective batch size is set to 4096 for all datasets except FGVC Aircraft, where it is reduced to 512 due to the smaller dataset size. Data augmentation follows standard PyTorch (Paszke et al., 2019) image pre-processing: RandomResizedCrop, horizontal flipping, and normalization. For vision-language models, we adopt their official preprocessing pipelines (e.g., OpenCLIP (Ilharco et al., 2021) transforms for CLIP, SigLIP, and SigLIP2) to ensure alignment with pre-training distributions. Experiments are conducted on a cluster of 8 NVIDIA A100 GPUs (40 GB VRAM each) and on the LUMI supercomputer, on clusters of 8 AMD Instinct MI250X GPUs (128 GB HBM2e memory each).

## C.2 EXPERIMENTAL RESULTS

**Accuracy vs. parameters.** We extend the benchmark presented on subsection 4.2 to include additional pre-training methods and datasets. Figure 9 and Figure 10 present the trade-off between top-1 accuracy and the number of trainable parameters (including the classifier) for various pooling/probing methods integrated into MAE and SimMIM with different backbone sizes (ViT-S, ViT-B and ViT-L). The evaluation spans multiple datasets, including FGVC-Aircraft, CUB200, Places365, CIFAR-100 and Cars196.

As shown in Figure 9a and Figure 9b, EP consistently outperforms standard linear probing across MAE ViT-S and ViT-L. Notably, on MAE ViT-L, $EP_{16}$ with $D_o = D_o/2$ achieves an accuracy boost of 79.1% surpassing linear probing by 3.1% while maintaining the same number of trainable parameters. Furthermore, $EP_{128}$ reaches 79.4%, outperforming SigLIP, while reducing the number of trainable parameters by over 11M.

In Figure 9c, we benchmark attentive probing on SimMIM ViT-B pre-trained on ImageNet-1K. Baselines such as [CLS] and GAP remain parameter-efficient but yield relatively low accuracy. Classical attention modules like CBAM and AbMILP do not improve this trade-off, while methods such as ViT, V-JEPA, and SigLIP achieve higher accuracy but at the cost of orders-of-magnitude more parameters. EP strikes a favorable balance: scaling the number of queries ($EP_2$–$EP_{64}$) consistently increases accuracy, while reducing $D_o$ effectively lowers parameter count with only moderate drops in performance. Notably, $EP_2$ with $D_o = D_i/2$ achieves 60.6% top-1 accuracy using fewer than 0.7M parameters, outperforming GAP and DELF under similar budgets. At the high end, $EP_{64}$ reaches 65.1% accuracy, closing the gap to heavy-weight probing methods while remaining lighter.

In Figure 9d, our $EP_{24}$ variant, for the FGVC-Aircraft dataset, achieves a remarkable accuracy boost of 61.2% (+19.5%), while maintaining lower parameter count than linear probing (41.7%). Similarly, in Figure 9e for the CUB200 dataset, our $EP_{64}$ with $D_o = D_o/4$ variant achieves comparable accuracy (75.9%) with computationally costly poolings such as SigLIP (77.8%) with around 7M trainable parameters less.

Finally, in Figure 9f, GAP and [CLS], the two primary baselines, exhibit high parameter efficiency but low classification accuracy. In contrast, methods like SigLIP (54.39%), ViT (54.84%), and V-JEPA (52.44%) achieve higher accuracy, albeit at the cost of increasing the number of trainable parameters. EP has the best trade-off between accuracy and parameters, achieving top-1 classification accuracy of 53.7% with just 1M extra trainable parameters ($EP_{64}$).

Moving to the additional datasets shown in Figure 10, we observe consistent benefits of EP. On CIFAR-100 (Figure 10a), $EP_{128}$ achieves 78.9%, which is close to the best-performing attention pooling methods, while using significantly fewer parameters. On Cars196 (Figure 10b), our $EP_{64}$ variant achieves 82.7% top-1 accuracy, clearly surpassing repurposed pooling methods such as DELF, while requiring far fewer parameters. Smaller variants (e.g., $EP_{16}$ and $EP_{24}$) already provide substantial gains over linear probing, showing that even lightweight configurations of our method maintain strong performance on fine-grained datasets.

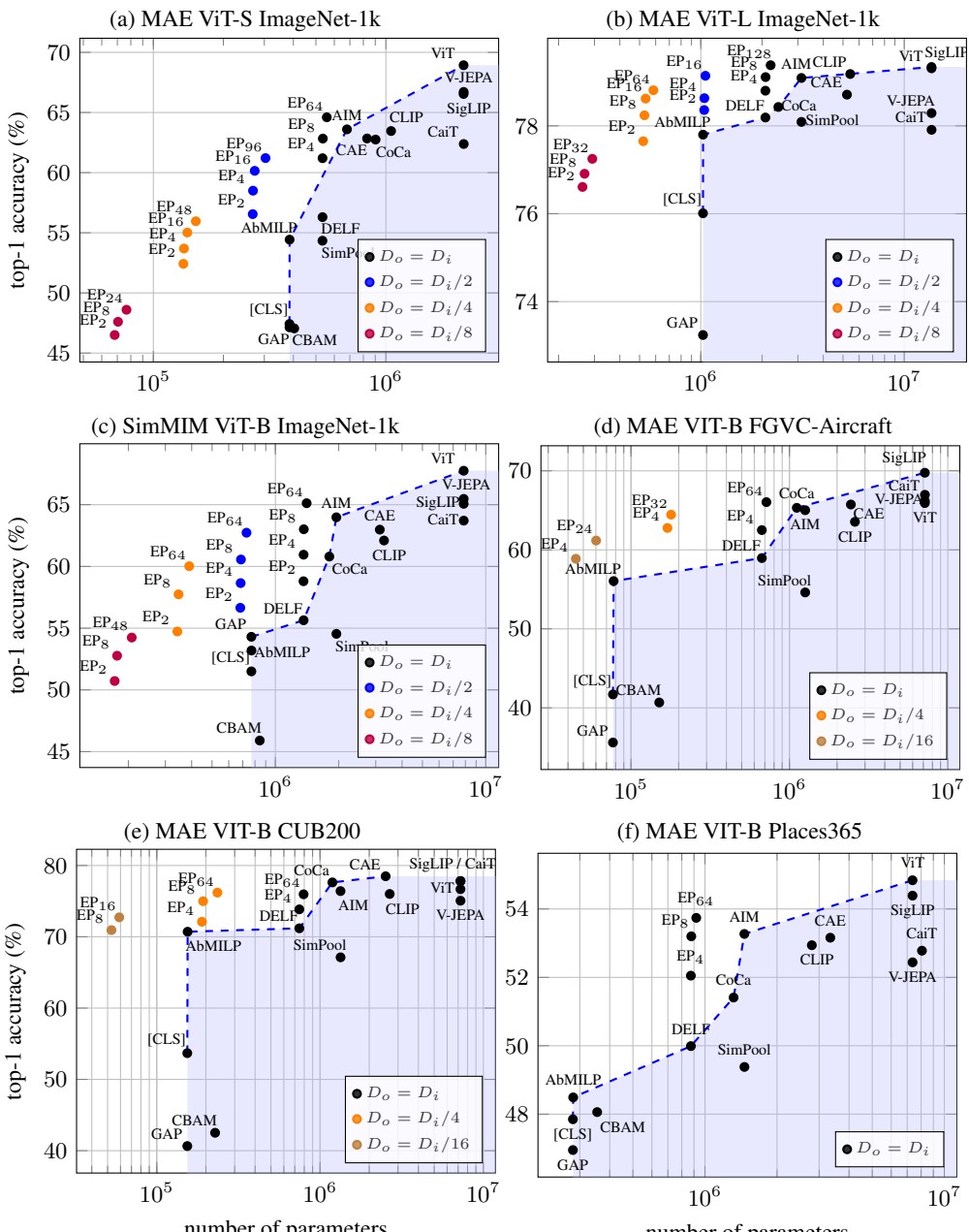

Figure 9: *Top-1 classification accuracy vs. number of trainable parameters* (including the classifier) for two self-supervised learning methods, with backbones of varying size (a, b) and across various datasets (d, e, f). EP variants are marked with different colors for different output dimensionalities $D_o$. $EP_M$: efficient probing with $M$ learnable queries. [CLS]: linear probing using the class token; GAP: global average pooling over patch tokens; ViT: default transformer block.

**Layer-wise probing.** Figure 11 presents a layer-wise comparison between standard linear probing (LP) and efficient probing (EP) using patch token representations from intermediate layers of a pre-trained and frozen MAE with ViT-B. While LP exhibits a clear degradation in performance as we move toward earlier layers (dropping from 67.7% at layer 12 to just 45.8% at layer 6), EP demonstrates remarkable robustness. It maintains high accuracy even from lower layers, with performance stabi-

Figure 11: *ImageNet-1K accuracy from intermediate MAE ViT-B layers* for LP vs. EP, with EP–LP gains.

| LAYER | LP | EP | GAIN |
|---|---|---|---|
| 12 | **67.7** | 75.6 | +7.9 |
| 10 | 66.2 | **75.9** | +9.7 |
| 9 | 64.5 | 75.4 | +10.9 |
| 6 | 45.8 | 69.6 | **+23.8** |

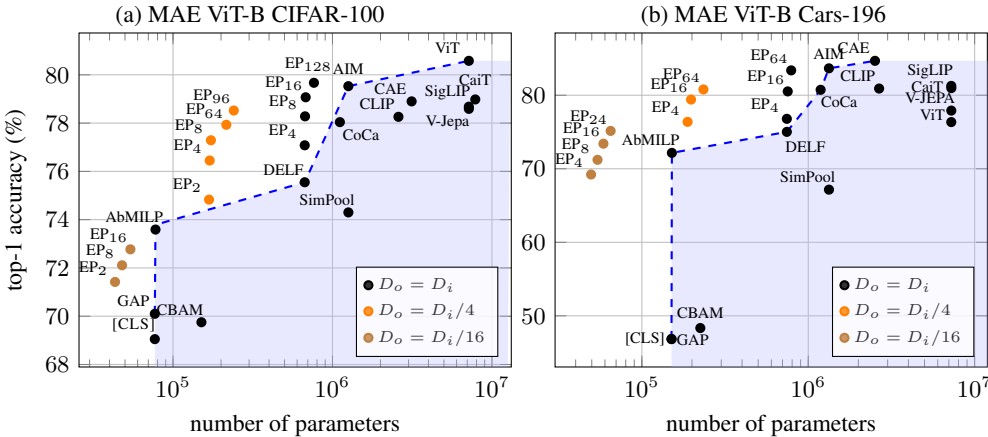

Figure 10: *Top-1 classification accuracy vs. number of parameters* for MAE ViT-B on two datasets (a, b). We evaluate both dedicated probing mechanisms (e.g., V-JEPA) and repurposed attentive pooling methods (e.g., CLIP). EP variants are marked with different colors for different output dimensionalities $D_o$. $EP_M$: efficient probing with $M$ learnable queries. [CLS]: linear probing using the classification token; GAP: global average pooling over patch tokens; VIT: default transformer block.

lizing beyond layer 9. Notably, EP yields a significant relative improvement of +23.8% at layer 6 over LP, underscoring its ability to extract and utilize meaningful representations from less semantically enriched stages of the encoder. These results highlight the effectiveness of EP in unlocking information from earlier layers that standard LP fails to exploit.

**Low-shot probing.** Table 4 evaluates the performance of LP, EP, and FT under limited supervision, using only 5% and 10% of the ImageNet-1K training set, stratified by class. Although LP struggles in this low-shot regime, EP substantially bridges the gap toward FT. Specifically, EP closes 74.8% and 71.5% of the LP→FT performance gap for the 5% and 10% subsets, respectively. These improvements are particularly impressive given that EP remains significantly more parameter-efficient than FT, with a complexity comparable to that of LP. These findings highlight the strong data efficiency of EP.

Table 4: *Top-1 accuracy on ImageNet-1k with limited training data for MAE ViT-B*. Results for linear probing (LP), efficient probing (EP), and fine-tuning (FT) on 5% and 10% subsets. The last column shows the percentage of the LP→FT performance gap closed by EP. For reference, the gap closed by EP on the full training set (100%) is 49.7%.

| SUBSET | LP | EP | FT | % GAP | |
|---|---|---|---|---|---|
| 5% | 49.6 | 60.9 | 64.7 | ▮▮▮▮▮▮ | 74.8% |
| 10% | 55.9 | 65.2 | 68.9 | ▮▮▮▮▮▮ | 71.5% |

**In- and out-of-domain k-NN evaluation.** To further examine how EP behaves relative to LoRA-based fine-tuning, we perform a cross-dataset k-NN evaluation on MAE ViT-B features, using ImageNet-1K, StanfordCars, and Food101 as target datasets (Table 5). The first row reports the baseline k-NN accuracy obtained directly from the frozen MAE backbone. The next three rows evaluate features produced by EP when the probe is trained on each dataset independently; the diagonal entries (e.g., 70.5% on ImageNet-1K, 70.0% on StanfordCars, 75.2% on Food101) correspond to in-domain performance, while off-diagonal entries measure cross-dataset generalization. The last three rows report the corresponding results when MAE is adapted with the best-performing LoRA configuration for each dataset. As expected, LoRA generally achieves the strongest in-domain accuracy (e.g., 72.3% on ImageNet-1K, 75.4% on StanfordCars, 80.3% on Food101), reflecting the benefit of supervised feature adaptation. However, EP consistently provides stronger or comparable out-of-domain performance: for instance, when trained on StanfordCars, EP features achieve 51.7% on ImageNet-1K and 42.0% on Food101, compared to 45.8% and 38.7% for LoRA-tuned features; similarly, EP trained on Food101 yields 58.2% on ImageNet-1K and 23.4% on StanfordCars, versus 56.7% and 15.2% for LoRA. These results suggest a complementary behavior: LoRA excels

at specializing the backbone to a specific task, while EP preserves more of the original pre-trained structure and thus offers more robust cross-dataset generalization.

Table 5: *Cross-dataset k-NN evaluation on MAE ViT-B using frozen, EP-probed, and LoRA-tuned features.* Default EP corresponds to EP$_{32}$, while the "best LoRA" configuration applies LoRA to all 12 layers on the $W_Q, W_K, W_V$, and $W_O$ projection matrices with rank $\rho$=8.

| FEATURES | k-NN EVALUATION ON | | |
|---|---|---|---|
| | IMAGENET-1K | STANFORDCARS | FOOD101 |
| frozen MAE backbone | 46.1 | 9.5 | 28.8 |
| + EP probed on ImageNet-1K | 70.5 | **31.4** | 64.2 |
| + EP probed on StanfordCars | **51.7** | 70.0 | **42.0** |
| + EP probed on Food-101 | 58.2 | 23.4 | 75.2 |
| + best LoRA tuned on ImageNet-1K | **72.3** | 24.7 | **65.6** |
| + best LoRA tuned on StanfordCars | 45.8 | **75.4** | 38.7 |
| + best LoRA tuned on Food-101 | 56.7 | 15.2 | **80.3** |

**Object localization.** A key empirical property of EP is that its queries specialize in *complementary* and semantically meaningful object parts (subsection 4.3). To examine EP utility beyond standard classification, we evaluate its performance in an unsupervised object localization setting, on ImageNet-1K (Table 6), using the WSOL protocol (Choe et al., 2020; 2022). We average EP 's attention maps, without any modification or additional training, and compare them to the standard last-layer [CLS] → patch attention. Across five backbones and three model sizes, EP consistently improves MaxBoxAccV2 by +9.8% on average. This demonstrates that EP 's attention maps act as strong unsupervised localizers "out-of-the-box".

Table 6: *Unsupervised ImageNet-1K localization* (MaxBoxAccV2). EP substantially improves localization quality across five backbones and three model sizes, with an average gain of +9.8% over baseline attention.

| MODEL | ARCH | LP | EP | $\Delta$ACC. (%) |
|---|---|---|---|---|
| MAE | ViT-S/16 | 46.0 | 58.5 | +12.5 |
| MAE | ViT-B/16 | 54.2 | 60.4 | +6.2 |
| MAE | ViT-L/16 | 46.9 | 61.2 | +14.3 |
| BEiTv2 | ViT-B/16 | 47.0 | 61.2 | +14.2 |
| SimMIM | ViT-B/16 | 45.2 | 60.0 | +14.8 |
| iBOT | ViT-B/16 | 57.6 | 63.7 | +6.1 |
| SigLIP | ViT-L/16 | 44.0 | 44.2 | +0.2 |
| AVERAGE | | 48.7 | **58.5** | +9.8 |

**Image retrieval.** Table 7 and Table 8 evaluate EP in a zero-shot image retrieval setting, following the standard Recall@K protocol on two fine-grained datasets, CUB200 and Cars196, respectively. In both cases, we use the same EP mechanism trained once on ImageNet-1K, without any further adaptation on the target dataset. Across all retrieval experiments, we utilize the entire datasets (full set of images and classes), all of which remain fully unseen during the training stage on ImageNet-1K. Every image serves as a query, ensuring a robust zero-shot evaluation protocol. On CUB200, EP consistently improves performance across five models (e.g., +40.8% for MAE, +11.2% for BEiTv2, +10.3% for iBOT, +5.3% for CLIP, +15.9% for SigLIP on R@1 metric). Similarly, on Cars196 the gains remain significant in most cases (e.g., +21.5% for MAE, +8.6% for iBOT, +4.6% for SigLIP). These findings show that EP 's features can be used "out-of-the-box" for image retrieval, consistently outperforming the features of the frozen backbones.

## C.3 EXPERIMENTAL ANALYSIS

**Attention complementarity.** To further expand our analysis of attention complementarity, we complement the average-based metric reported in Figure 7 with an additional measure: 1− max off-diagonal similarity, which reflects the strongest redundancy between predictors. As shown in Figure 12a and Figure 12b, EP achieves substantially higher complementarity than the internal MHSA heads across all backbones, regardless of whether we use the average or max metric. A natural concern is whether this effect is biased by considering only the last block. To address this, we compute complementarity for MAE ViT-B across all 12 blocks (Figure 12c). The best scores obtained internally (0.37 for average, 0.10 for max) remain far below those of EP (0.65 and 0.22, respectively), confirming that the gap is not specific to the last block. Another question is whether the low diversity arises from self-attention itself. However, even SigLIP—which uses a cross-attention mechanism in

Table 7: *Zero-shot image retrieval performance (Recall@K) on CUB200 dataset (images: 11788, classes: 200)* across five different backbones with and without Efficient Probing (EP).

| FEATURES | R@1 | R@2 | R@4 | R@8 | R@16 | R@32 |
|---|---|---|---|---|---|---|
| frozen MAE backbone | 16.3 | 23.7 | 33.3 | 45.0 | 58.0 | 70.8 |
| + EP on ImageNet-1K | 57.1 | 69.5 | 79.5 | 87.5 | 93.0 | 96.2 |
| frozen BEiTv2 backbone | 57.0 | 68.2 | 78.0 | 85.8 | 90.9 | 94.2 |
| + EP on ImageNet-1K | 68.2 | 78.7 | 86.7 | 92.3 | 95.9 | 97.8 |
| frozen iBOT backbone | 51.8 | 64.0 | 74.7 | 83.2 | 90.0 | 94.1 |
| + EP on ImageNet-1K | 62.1 | 73.7 | 82.4 | 89.7 | 94.0 | 96.8 |
| frozen CLIP backbone | 75.0 | 84.3 | 91.3 | 95.3 | 97.5 | 98.7 |
| + EP on ImageNet-1K | 80.3 | 88.0 | 92.9 | 95.8 | 97.6 | 98.6 |
| frozen SigLIP backbone | 60.8 | 72.1 | 81.6 | 88.8 | 93.5 | 96.4 |
| + EP on ImageNet-1K | 76.7 | 85.7 | 92.0 | 95.7 | 97.8 | 98.8 |

Table 8: *Zero-shot image retrieval performance (Recall@K) on Cars196 dataset (images: 16185, classes: 196)* across five different backbones with and without Efficient Probing (EP).

| FEATURES | R@1 | R@2 | R@4 | R@8 | R@16 | R@32 |
|---|---|---|---|---|---|---|
| frozen MAE backbone | 12.7 | 17.6 | 23.7 | 31.8 | 42.9 | 55.7 |
| + EP on ImageNet-1K | 34.2 | 44.8 | 55.6 | 65.9 | 76.4 | 85.2 |
| frozen BEiTv2 backbone | 48.3 | 60.7 | 72.3 | 81.8 | 89.2 | 94.5 |
| + EP on ImageNet-1K | 44.2 | 56.4 | 68.5 | 79.1 | 87.3 | 93.7 |
| frozen iBOT backbone | 31.5 | 40.7 | 50.2 | 60.0 | 69.9 | 79.0 |
| + EP on ImageNet-1K | 40.1 | 50.6 | 61.0 | 70.7 | 79.5 | 87.6 |
| frozen CLIP backbone | 79.6 | 88.9 | 94.6 | 97.8 | 99.2 | 99.7 |
| + EP on ImageNet-1K | 79.8 | 89.3 | 94.8 | 97.8 | 99.2 | 99.7 |
| frozen SigLIP backbone | 85.8 | 92.3 | 96.1 | 98.0 | 99.1 | 99.6 |
| + EP on ImageNet-1K | 90.4 | 95.4 | 97.8 | 99.0 | 99.6 | 99.8 |

its last block—still yields lower complementarity than EP, suggesting that the effect is not explained by self-attention alone.

Figure 12d and Figure 12e further compare EP against other attentive probing methods (V-JEPA, AIM). Interestingly, all attentive probing mechanisms achieve higher complementarity than the internal MHSA heads, with AIM coming close to EP. This pattern might suggest that attentive probing, when designed effectively, encourages predictors to specialize in complementary regions, likely because probing operates on frozen backbones and must learn to aggregate features as efficiently as possible. This evidence indicates that complementarity may be an inherent property of attentive probing, rather than a byproduct of backbone architecture or attention type, opening new directions for future work.

**Attention complementarity as a loss.** We further examine the role of query complementarity by encouraging diversity between queries/heads via an auxiliary attention-similarity loss. Specifically, for AIM, V-JEPA, and EP we add to the cross-entropy loss, an extra term $\mathcal{L}_{attn}$ that penalizes similarity between the attention maps of different queries. On MAE ViT-B probed on ImageNet-1K, this yields a small gain for V-JEPA ($74.1\% \rightarrow 74.3\%$), while AIM and EP retain essentially identical final accuracy but converge faster (e.g. EP reaches $64.1\%$ at epoch 10 with $\mathcal{L}_{attn}$ vs. $60.0\%$ without it, before both converge to the same final score). These results suggest that complementarity is a beneficial property that can modestly help or accelerate training. However, methods such as AIM and EP, learn highly

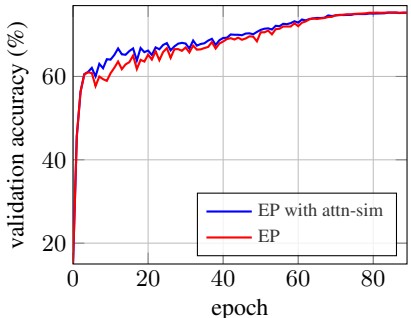

Figure 13: *Validation top-1 accuracy on ImageNet-1K over training epochs for EP, with and without the auxiliary attention-similarity loss.* The additional loss slightly accelerates early training but both variants converge to identical final performance.

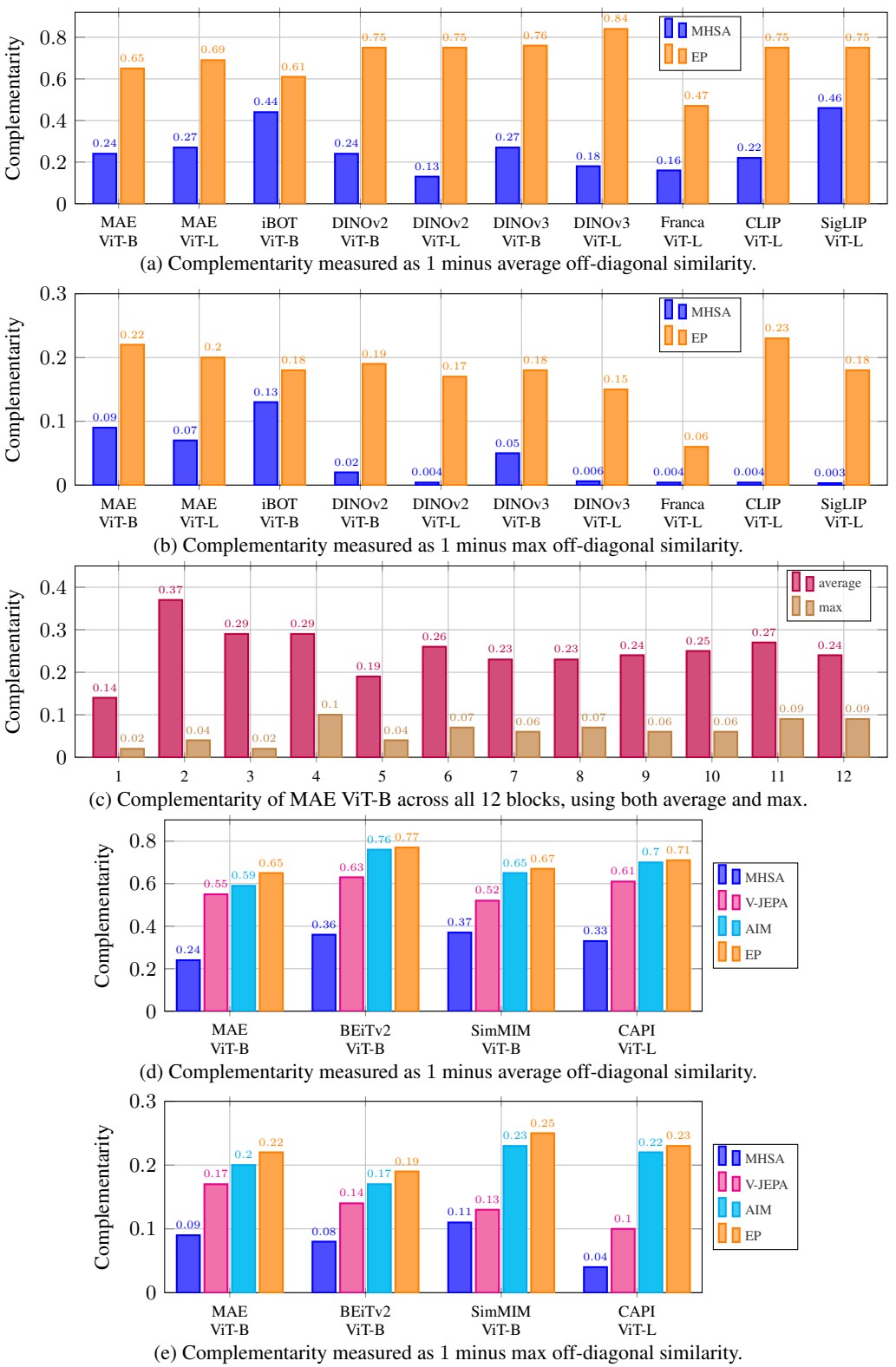

Figure 12: *Complementarity scores of attention maps across different backbones (a, b, c) and probing methods (d, e).* We compare the diversity of internal MHSA heads in the last block against the external V-JEPA heads, AIM heads, and EP queries.

complementary queries, without $\mathcal{L}_{\text{attn}}$, via the attention mechanism that naturally discovers this structure on its own.

**Entropy.** Figure 14 examines the relation between last block attention entropy and probing performance across different pre-training methods. Models with lower entropy exhibit more concentrated and focused attention distributions, correlating with stronger probing accuracy under EP. This is particularly evident for methods like as DINOv2 and DINOv3, which couple low entropy with strong LP and EP performance. In contrast, models like MAE ViT-S (MAE-S) or SimMIM ViT-B (SimMIM-B) show higher entropy, reflecting more diffuse attention and correspondingly weaker probing under LP. Crucially, EP consistently boosts accuracy across all entropy levels, with bubble sizes indicating particularly large gains for the high-entropy models. This aligns with our broader finding that methods optimizing patch-level representations rather than explicit global tokens benefit most from attentive probing, as EP effectively compensates for diffuse attention and makes probing more robust to the quality of backbone distributions.

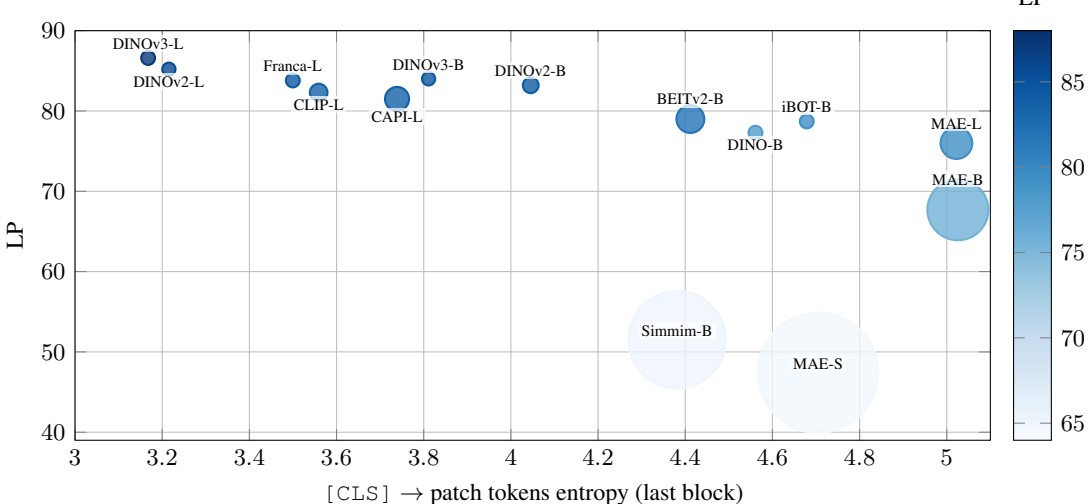

Figure 14: *Entropy analysis of last-block attention distribution across different pre-training methods.* Bubble color indicates efficient probing (EP) accuracy, bubble size encodes $\Delta$ accuracy (EP−LP). Lower entropy corresponds to more focused attention and higher accuracy under EP.

**Impact of attention predictors, $D_a$, and $D_o$.** We analyze the effect of increasing the number $M$ of heads (in AIM) and the number $M$ of queries (in EP) on probing performance. Figure 15 shows that both lead to accuracy improvements. For AIM, increasing the number of heads incurs no additional cost in terms of parameters, but its effectiveness depends on the presence of $W_K$. In contrast, EP achieves similar or better performance by leveraging additional queries, while removing $W_K$. AIM introduces an additional attention dimensionality $D_a$, since its query is learnable and interacts with $W_K$. Lowering $D_a$ reduces the parameters but leads to a greater accuracy drop (green points), indicating that the learned query formulation benefits from a large attention space. We also evaluate the impact of reducing the output dimensionality $D_o$ (blue points). On EP, we observe that lowering $D_o$ to $D_i/2$ reduces

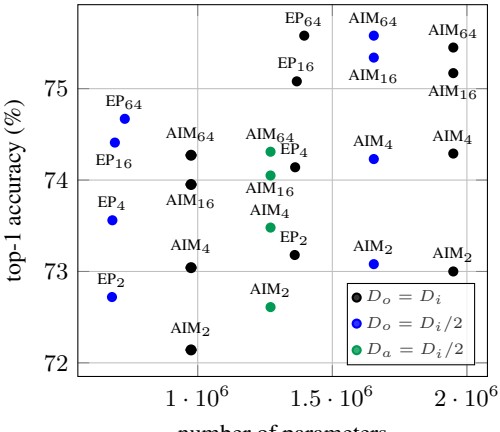

Figure 15: *Effect of varying the number of heads/queries $M$, $D_a$, and output dimension $D_o$ on probing accuracy.* Black: standard ($D_o = D_i$); blue: reduced classifier dimension ($D_o = D_i/2$); green: reduced attention dimension ($D_a = D_i/2$); half-blue-half-green: simultaneous reduction of both $D_o$ and $D_a$.

parameters while maintaining competitive performance. Interestingly, this strategy also generalizes well to AIM, demonstrating that extracting lower-dimensional features can achieve comparable accuracy with reduced computational cost.

**EP convergence.** Although all probing methods are trained for 90 epochs for fairness (following the protocol of He et al. (2022) and Przewięźlikowski et al. (2025)), we also examine how quickly EP converges. As shown in Table 9, EP trained for only 10 epochs (EP@10) already matches or even surpasses the performance of standard linear probing trained for the full 90 epochs (LP@90) across 12 models. Moreover, EP recovers on average more than 97% of its final accuracy within these first 10 epochs. These results highlight that in practical scenarios, EP requires only a small fraction of the standard training budget to achieve near-optimal probing performance and produce highly discriminative features.

Table 9: *EP convergence on ImageNet-1K.* EP@10 is EP accuracy after 10 epochs; LP@90 and EP@90 are final probing results after 90 epochs.

| MODEL | EP@10 | LP@90 | EP@90 | EP@10 VS. LP@90 GAP | EP@10 VS. EP@90 GAP |
|---|---|---|---|---|---|
| MAE ViT-B/16 | 60.0 | 67.7 | 75.6 | 88.6% | 79.4% |
| BEiTv2 ViT-B/16 | 80.0 | 79.0 | 81.7 | 101.3% | 97.9% |
| CAPI ViT-L/14 | 82.3 | 81.5 | 83.6 | 101.0% | 98.4% |
| iBOT ViT-B/16 | 78.6 | 78.7 | 79.2 | 99.9% | 99.2% |
| DINOv2 ViT-B/14 | 83.6 | 83.2 | 84.0 | 100.5% | 99.5% |
| DINOv2 ViT-L/14 | 85.0 | 85.2 | 85.6 | 99.8% | 99.3% |
| Franca ViT-B/16 | 83.8 | 83.8 | 84.3 | 100.0% | 99.4% |
| DINOv3 ViT-B/16 | 84.4 | 84.0 | 84.4 | 100.5% | 100.0% |
| DINOv3 ViT-L/14 | 86.7 | 86.6 | 87.1 | 100.1% | 99.5% |
| CLIP ViT-L/14 | 82.7 | 82.3 | 83.4 | 100.5% | 99.2% |
| SigLIP ViT-L/16 | 85.6 | 84.1 | 86.1 | 101.8% | 99.4% |
| AIMv2 ViT-L/14 | 85.5 | 84.8 | 85.9 | 100.8% | 99.5% |

**Attention mixing.** We further explore an attention mixing variant of EP, where a larger set of query maps is linearly projected into a smaller number of effective maps. For example, projecting 64 queries into 8 maps achieves 75.6% accuracy, outperforming the 8-query baseline (74.8%) and suggesting that mixing can improve the expressivity of a small query budget. However, compared to the un-projected 32 or 64 query variants (75.6%), attention mixing offers no clear advantage while introducing additional parameters through the projection layer. For this reason, we do not include attention mixing in the final design of EP, though it may inspire future extensions exploring alternative aggregation strategies.

Table 10: *Comparison of* EP *variants using different numbers of original maps (queries) and projected maps on ImageNet-1K with MAE ViT-B.* Attention mixing refers to projecting multiple query maps into a smaller set via a linear+softmax layer. While projection improves over directly training with few queries (e.g., 64→8 vs. 8 original), it does not outperform the un-projected setting with the same number of queries, and adds extra parameters.

| # ORIGINAL EP MAPS | # PROJECTED EP MAPS | ACCURACY | # ORIGINAL EP MAPS | ACCURACY |
|---|---|---|---|---|
| 32 | 8 | 75.4 | | |
| 64 | 8 | **75.6** | 8 | 74.8 |
| 128 | 8 | 75.5 | | |
| 64 | 16 | 75.5 | | |
| 128 | 16 | **75.8** | 16 | 75.0 |
| 256 | 16 | 75.6 | | |
| 64 | 32 | 75.6 | | |
| 128 | 32 | 75.6 | 32 | 75.6 |
| 256 | 32 | **75.7** | | |
| 128 | 64 | 75.8 | | |
| 256 | 64 | 75.8 | 64 | 75.6 |
| 384 | 64 | **75.9** | | |
| 256 | 128 | 75.8 | | |
| 384 | 128 | **76.0** | 128 | 75.5 |

**Matryoshka representation learning.** In our accuracy vs. parameter trade-offs (e.g., Figure 2), each evaluation scale ($D_o = D_i, D_i/2, D_i/4, D_i/8$) requires training a separate classifier, and the resulting accuracies potentially serve as an *upper bound* for that dimensionality, since each probe is fully specialized to its target output size. To reduce training cost and enable a single probe that operates across multiple scales, we investigate whether *Matryoshka representation learning* (Kusupati

et al., 2022)—originally proposed for training from scratch or fine-tuning—can be applied to attentive probing under EP. Matryoshka jointly optimizes probing across multiple output dimensionalities by summing their losses (e.g., $L=\lambda L_D + L_{D/2}$). We study two variants: (i) Efficient Matryoshka, which uses a single classifier with nested subspaces spanning multiple dimensionalities, and thus adds no extra parameters; (ii) Vanilla Matryoshka, which trains separate classifiers for each dimensionality, increasing the parameter count. Table 11 reports results with MAE ViT-B probed using EP. Without Matryoshka, probing at $D$ ($D_o=D_i$) achieves the highest accuracy (75.6%), but performance drops sharply at smaller dimensions (50.7% at $D/2$, 20.1% at $D/4$, 5.5% at $D/8$). Efficient Matryoshka substantially improves performance at reduced dimensions (e.g., +22.5% at $D/2$) while maintaining accuracy at $D$. Accuracy at $D$ slightly decreases as $\lambda$ (the weight of the full-dimensional loss) is reduced, reflecting the trade-off between full- and low-dimensional performance. Vanilla Matryoshka alleviates the performance drop at $D$ but requires more parameters. To further illustrate these effects, we evaluate the standard linear probing (LP) baseline without Matryoshka under the same $D/2$, $D/4$, $D/8$ evaluation protocol, as also extending LP with Matryoshka. In Table 12 we observe that standard LP (without Matryoshka) suffers from an even sharper performance collapse than EP (e.g., 34.5% at $D/2$, 10.8% at $D/4$, and 1.4% at $D/8$). This confirms that the drop at reduced dimensions is not specific to EP, but is an inherent limitation of evaluating ViT representations at truncated dimensionalities without explicit multi-scale optimization. Overall, Matryoshka probing provides a promising way to probe across multiple scales simultaneously, improving efficiency, and adaptability of evaluation. However, it does not yet match the upper-bound performance of dimension-specific probes, thus we present it here as an exploratory analysis that highlights an exciting direction for future work.

Table 11: *Comparison of* EP *with and without Matryoshka representation learning* on ImageNet-1K (MAE ViT-B). Without Matryoshka, probing at $D$ ($D_o=D_i$) achieves the best accuracy but collapses at smaller dimensions. Efficient Matryoshka enables multi-scale probing without extra parameters, improving performance at reduced dimensions with relatively small degradation at $D$. Vanilla Matryoshka partly restores performance at $D$ but at the cost of additional parameters.

| Eval. on | Without Matryoshka | | | | Efficient Matryoshka | | | | Vanilla Matryoshka | |
| | $D$ | $D/2$ | $D/4$ | $D/8$ | $D, D/2, D/4, D/8$ | | | | $D, D/2, D/4, D/8$ | |
| | | | | | $\lambda=0.8$ | $\lambda=0.6$ | $\lambda=0.4$ | $\lambda=0.25$ | $\lambda=0.6$ | $\lambda=0.25$ |
|---|---|---|---|---|---|---|---|---|---|---|
| $D$ | **75.6** | – | – | – | 75.0 | 74.7 | 74.3 | 73.9 | 75.4 | 75.2 |
| $D/2$ | 50.7 | **74.4** | – | – | 72.2 | 73.1 | 73.6 | 73.8 | 73.8 | 74.4 |
| $D/4$ | 20.1 | – | **72.4** | – | 69.5 | 70.9 | 71.4 | 71.6 | 71.7 | 72.2 |
| $D/8$ | 5.5 | – | – | **69.8** | 65.7 | 67.3 | 67.9 | 68.2 | 68.6 | 68.9 |
| # Par. | 1.4M | 1.4M | 1.4M | 1.4M | 1.4M | | | | 2.1M | |
| | 5.5M | | | | | | | | | |

Table 12: *Comparison of LP with and without Matryoshka representation learning* on ImageNet-1K (MAE ViT-B). Without Matryoshka, linear probing collapses sharply at reduced output dimensionalities. Matryoshka improves low-dimensional performance, without increasing parameters, though the accuracy at $D$ decreases as $\lambda$ (the weight of the full-dimensional loss) is reduced.

| Eval. on | Without Matryoshka | Efficient Matryoshka | | | |
| | | $\lambda = 0.8$ | $\lambda = 0.6$ | $\lambda = 0.4$ | $\lambda = 0.25$ |
|---|---|---|---|---|---|
| $D$ | **66.4** | 66.1 | 65.8 | 65.4 | 65.0 |
| $D/2$ | 34.5 | 59.2 | 60.6 | 61.2 | 61.4 |
| $D/4$ | 10.8 | 52.5 | 54.5 | 55.3 | 55.7 |
| $D/8$ | 1.4 | 43.4 | 46.1 | 47.0 | 47.4 |

**Visualizations.** To better understand the behavior of different attentive pooling/probing methods, we present visualizations of attention maps across various configurations.

Figure 16 explores the effect of varying the number of queries in EP, visualizing configurations with 1, 2, 4, and more queries. When only a single query ($EP_1$) is used, the attention map tends to capture a coarse, global representation of the object. As the number of queries increases, the attention becomes more fine-grained and spatially distributed, with each query specializing in distinct object

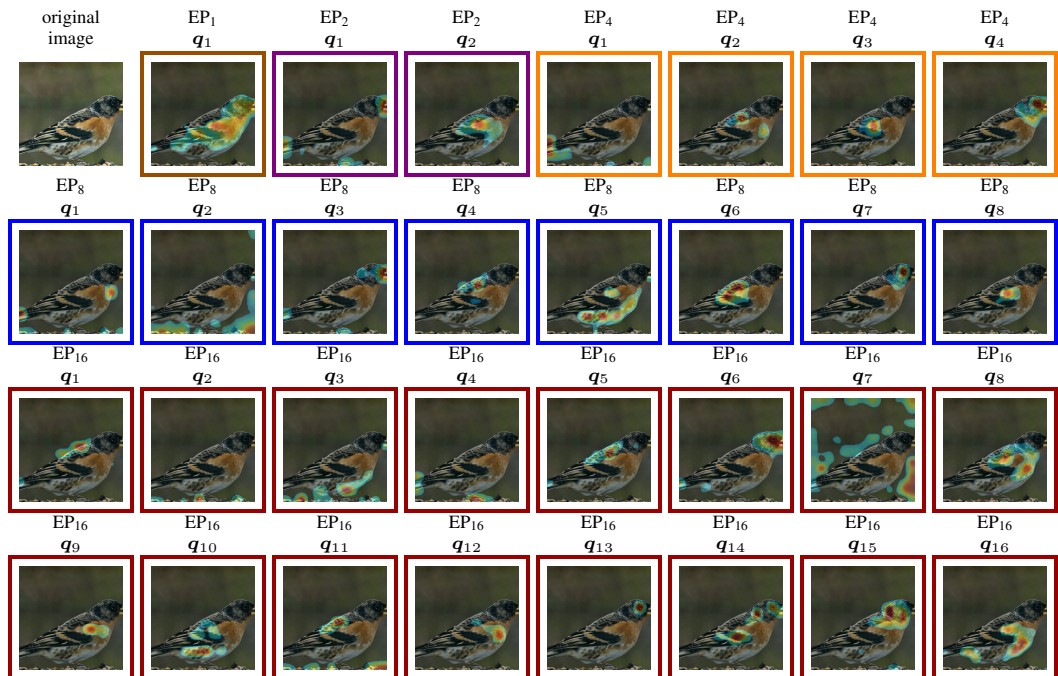

Figure 16: *Attention maps of efficient probing (*EP*) variants grouped by the number of queries (1, 2, etc.).* MAE ViT-B pre-trained on ImageNet-1K, probed with EP. Images: ImageNet-1k validation set.

regions. This highlights the flexibility of EP in controlling the granularity of attention: fewer queries encourage holistic coverage, while more queries promote detailed, part-based localization.

Figure 17 shows the attention maps obtained from four single-head attention probing methods (CBAM, AbMILP, DELF, and SimPool) using an ImageNet-1K pretrained MAE ViT-B model. Among them, CBAM exhibits poor localization, often failing to focus on the target object, which is consistent with its low classification accuracy across datasets. In contrast, AbMILP, DELF, and SimPool produce more precise and meaningful attention, highlighting relevant object regions while suppressing background noise. Due to their single-head nature, these methods are compelled to concentrate all semantic information into a single attention vector, which encourages a global view of the input image rather than fine-grained discrimination.

Figure 18 compares attention maps from multi-head probing methods. Rather than visualizing just the average attention across heads,which can obscure useful per-head behavior, we show the minimum, maximum, and standard deviation across attention heads. The first column contains maps from the [CLS] token of the pretrained MAE ViT-B model. The remaining columns display maps from CAE, CaiT, CLIP, CoCa, ViT, V-JEPA, SigLIP, and AIM, alongside EP using 16 learnable queries (EP$_{16}$). Notably, EP produces high-quality attention maps that rival the best-performing methods in both clarity and relevance, while retaining computational efficiency.

Figure 19 presents the attention maps corresponding to each individual query in EP. We observe that each query $q_i$ attends to distinct, complementary regions of the object (e.g., head, torso, boundaries), illustrating how EP distributes attention cooperatively across salient features without redundancy. This diversity among queries reveals the model's capacity to decompose complex objects into meaningful sub-parts.

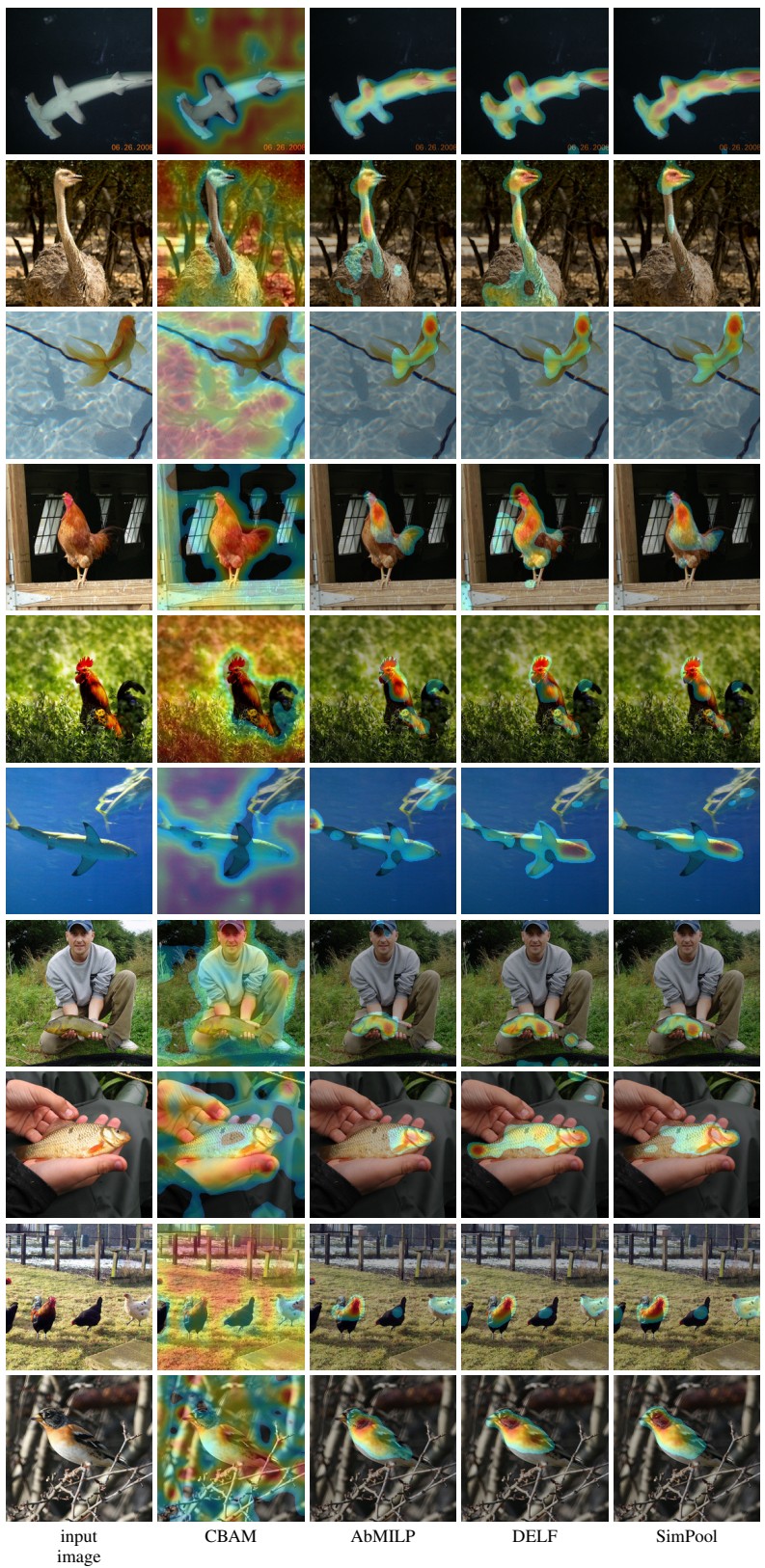

| input image | CBAM | AbMILP | DELF | SimPool |

Figure 17: *Attention maps of single-head attention pooling methods.* MAE ViT-B pre-trained on ImageNet-1k. Images: ImageNet-1k validation set.

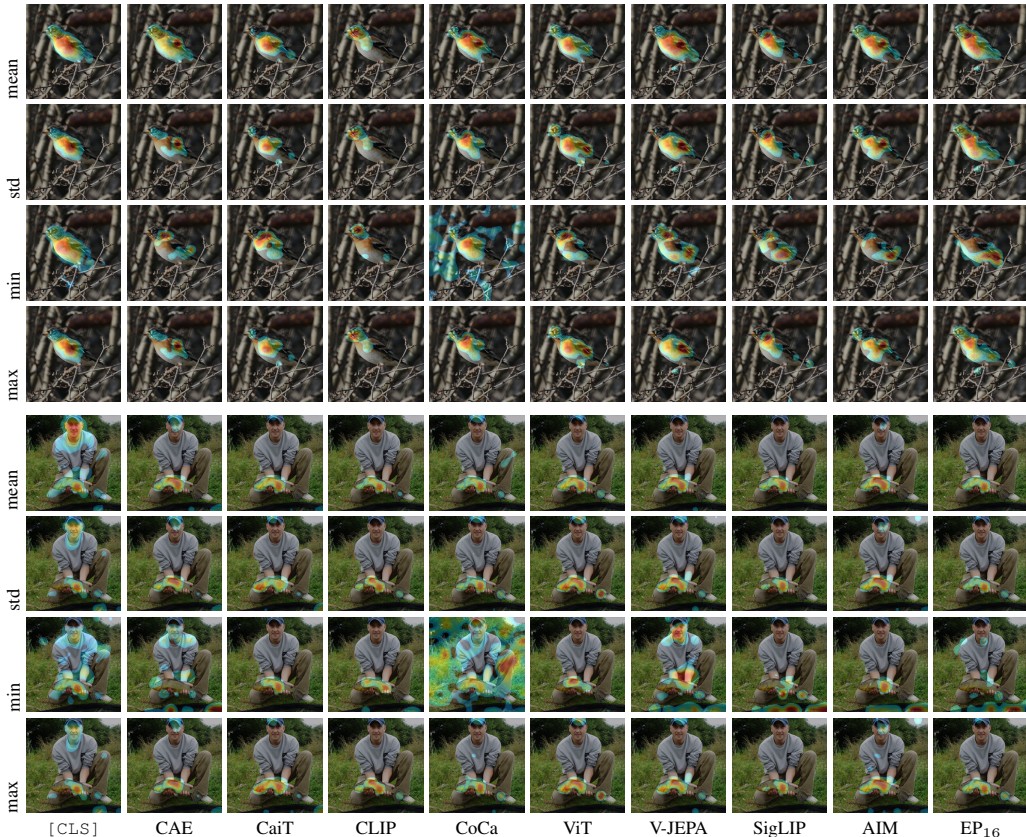

Figure 18: *Attention maps of multi-head attention pooling methods* for different attention predictor aggregators: mean, standard deviation (std), minimum (min), and maximum (max). MAE ViT-B pre-trained on ImageNet-1K. Images: ImageNet-1k validation set. $EP_{16}$: efficient probing (EP) with 16 queries.

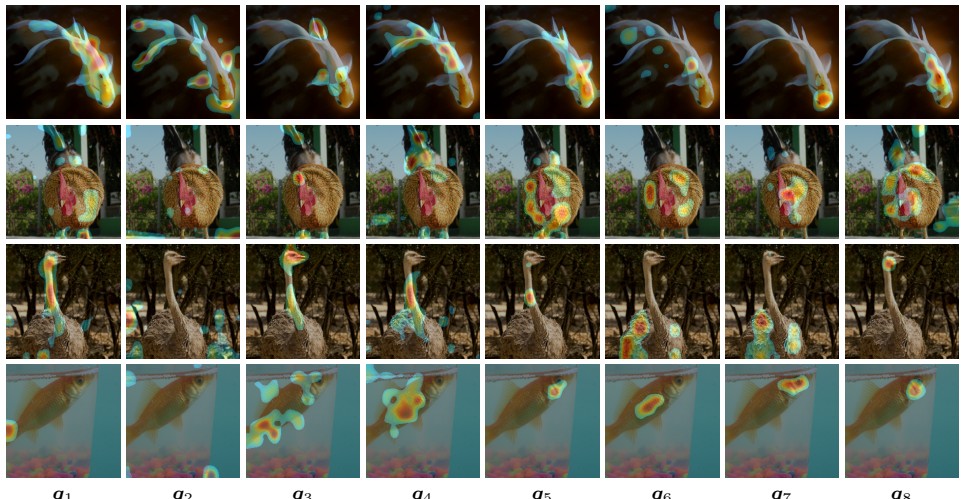

Figure 19: *Attention maps of efficient probing (EP)* with 8 queries. Each query $q_i$ learns to focus on distinct and complementary regions, capturing diverse spatial and semantic information. MAE ViT-B pre-trained on ImageNet-1K, probed with EP. Images: ImageNet-1k validation set.

# D  LIMITATIONS AND FUTURE WORK

While our study provides the first systematic benchmark of attentive probing and introduces a lightweight yet effective alternative, several limitations remain. First, our evaluation focuses exclusively on frozen backbones. Although this setting isolates the effect of probing mechanisms, it leaves open how efficient probing might interact with lightweight fine-tuning strategies or adapter-based methods. Exploring the synergy between probing and parameter-efficient fine-tuning could offer a broader view of scalable evaluation.

Second, our current experimental protocol always performs a full forward pass of the backbone during training, even though the backbone is frozen. In practice, one could pre-compute and store patch-token features once and run probing purely on top of these cached features. Systematically studying this "frozen-feature" protocol—its memory/IO trade-offs, impact on optimization dynamics, and applicability across architectures—is an interesting direction for future work.

Third, while we introduced complementarity metrics (average and max similarity) to study the diversity of attention predictors, our analysis remains largely diagnostic. A deeper theoretical understanding of why attentive probing tends to yield more complementary maps—and whether this property can be explicitly optimized—could further extend probing beyond evaluation into representation refinement.

Fourth, our experiments concentrate on image classification benchmarks. Attentive probing may also benefit other tasks that naturally require part-level reasoning, such as detection, segmentation, or retrieval, where its complementary attention maps could act as implicit part detectors. Extending probing to such structured tasks is a promising direction.

Fifth, although we examined variants such as attention mixing and Matryoshka probing, both were limited to ImageNet-scale experiments. Their potential on larger models and multimodal settings (e.g., vision–language tasks) remains underexplored. Similarly, the alternative protocol of probing with pre-stored frozen features, requiring only one backbone forward pass, is left for future work.

Finally, the broader implications of probing as more than an evaluation protocol deserve attention. Our results suggest that attentive probing exhibits emerging properties—such as diversity and interpretability—that are not trivially inherited from the backbone. Understanding whether these properties generalize across modalities and can be exploited for tasks like explainability, robustness, or adaptive computation opens up an exciting line of future research.

# E  USE OF LARGE LANGUAGE MODELS

This paper has made limited use of large language models (LLMs), specifically to aid in the polishing and refinement of writing. LLMs were not used for ideation, technical contributions, experimental design, analysis, or related work retrieval. All research ideas, methodology, experiments, and conclusions presented are solely the work of the authors.

