# OpenReview forum: "Attention, Please! Revisiting Attentive Probing Through the Lens of Efficiency"
_ICLR.cc/2026/Conference — ICLR 2026 Poster_

### Official Review · Reviewer_Vb6E · 2025-10-27

**Soundness:** 2
**Presentation:** 2
**Contribution:** 2
**Rating:** 2
**Confidence:** 2

**Summary:**

This paper addresses the limitations of standard linear probing (LP) for evaluating pre-trained visual representations, particularly for models like those based on Masked Image Modeling (MIM) that distribute information across patch tokens rather than relying on a single global representation (like a [CLS] token). The authors present a systematic study of attentive probing (AP), a protocol that uses attention to selectively aggregate these patch features. They introduce a novel, parameter-efficient AP method called Efficient Probing (EP), which uses a simple multi-query cross-attention mechanism without redundant projections. EP demonstrates state-of-the-art accuracy compared to LP and prior AP methods across various benchmarks and pre-training paradigms while being significantly more efficient in terms of parameters and computational cost. Furthermore, the analysis reveals that EP's high performance correlates with the localization quality of its attention maps, which are shown to be highly diverse and complementary.

**Strengths:**

1. The paper correctly identifies and addresses the misalignment between standard LP and modern pre-training paradigms (MIM, auto-regressive, diffusion) where discriminative information is distributed across patch tokens. Attentive probing is established as the necessary alternative.

2.  This is presented as the "first comprehensive study" of attentive probing, offering a unified framework that categorizes existing methods (including those from unrelated tasks) and thoroughly benchmarks their performance against parameter and computational efficiency.

3. The proposed Efficient Probing (EP) method is simple, yet highly effective. By reformulating the attention mechanism as a transformation-free multi-query cross-attention, it significantly improves the accuracy vs. parameter efficiency trade-off, consistently outperforming baselines and competitors.

4. The results show that EP provides consistent and substantial gains across diverse pre-training methods (MIM, JEA, Hybrid, VLM, Generative), with the largest gains seen in models whose pre-training optimizes patch tokens (e.g., DiT, SimMIM).

**Weaknesses:**

1. The paper positions its contribution against a backdrop where attentive probing is described as "underexplored" and existing methods "suffer from excessive parameterization and poor computational efficiency." While EP solves these issues, the baseline comparison set may be inherently weak due to the newness of the protocol, potentially overstating the relative accuracy gain compared to a hypothetical future efficient method.

2. The motivation for removing the key transformation ($W_{Kj}$) in EP is primarily efficiency. While the poor performance of a partially-shared-subspace design (Equation 8) is mentioned, the core reasoning for why learnable queries $u_j \in \mathbb{R}^{D_i}$ directly interacting with the full feature space $X$ (Equation 10) works better than projected queries $q_j \in \mathbb{R}^{d_a}$ interacting with projected keys $K_j$ (Equation 7) could be elaborated with a more formal argument about representation capacity or gradient flow beyond just empirical verification. The section on the impact of $W_K$ and $W_V$ in MHCA is incomplete in the provided text.

3. While the probing overhead (GFLOPs) is much smaller than the backbone, the overall evaluation still requires training for 90 epochs. The abstract/introduction states that full fine-tuning (FT) is "unsustainable and prohibitive at scale," but the computational cost of an attention-based protocol, even an efficient one, is still much higher than standard LP. This is mitigated but not entirely eliminated, especially when comparing against "frozen and pre-stored features" (mentioned in Figure 3 caption) which is a crucial future direction not implemented here.

4. The analysis is incomplete because it omits a comparison with crucial modern, low-cost alternatives that are widely used for large-scale model assessment, such as Zero-Shot classification for Vision-Language Models (VLMs) and PEFT techniques like LoRA. These methods represent highly relevant alternatives to EP's parameter efficiency and performance trade-off in the modern evaluation landscape.

**Questions:**

1. Can EP be adapted for non-classification tasks, and what modifications would be necessary for such applications?

2. How does EP scale when applied to very large backbone models or extremely large datasets, and are there cases where efficiency or accuracy gains diminish?

3. Considering that modern large models often use a vision backbone as a component in multimodal architectures, does the probing result truly reflect the backbone’s capacity, and is probing genuinely necessary in this context?

4. If probing is indeed necessary, would it be possible to supplement the analysis with comparisons to zero-shot and LoRA performance?

---

> ### Author Response · Authors · 2025-11-25
> **Answer to Reviewer Vb6E (part 1)**
>
> We thank **Reviewer Vb6E** for their review of our work. We appreciate the reviewer’s clear articulation of the **limitations of standard linear probing** for modern pre-training paradigms, and their **recognition that attentive probing provides a more suitable evaluation protocol** in these settings. We are grateful for the reviewer’s acknowledgement of the **novelty and value of our systematic study of attentive probing**, as well as their appreciation of the **simplicity** and **effectiveness** of Efficient Probing (EP) and its **strong performance–efficiency trade-offs**. We also thank the reviewer for highlighting the **breadth of our experiments** across diverse pre-training paradigms, and for noting the **interpretability insights** offered by our attention localization analysis. Finally, the reviewer's question regarding the relationship between PEFT and attentive probing *prompted us to extend our analysis to a broader comparison between PEFT and probing*—an angle we believe provides **meaningful value to the community**. We address the reviewer’s concerns point-by-point below.
>
> -----

---

> > ### Author Response · Authors · 2025-11-25
> > **Answer to Reviewer Vb6E (part 2)**
> >
> > **EP for non-classification tasks.**
> >
> > We thank the reviewer for this question. EP is **not tied to classification**—its mechanism is a set of learnable query vectors that extract information from patch tokens via a lightweight multi–query cross-attention. This makes EP directly applicable to any task where one needs to (i) *aggregate tokens*, or (ii) *select or weight spatial regions*. Below we show that EP can be successfully used for both **object localization** and **image retrieval**.
> >
> > ### EP for localization
> >
> > A key empirical property of EP is that its queries specialize in *complementary* and *semantically meaningful object parts* (subsection 4.3). This naturally enables non-classification applications. We evaluate EP for *unsupervised object localization* on ImageNet-1K using the WSOL protocol [1]. We simply average EP’s attention maps, without any modification or additional training, and compare them to the standard last-layer [CLS]→patch attention. Across 5 backbones and 3 model sizes, EP consistently improves MaxBoxAccV2 by **+9.8%** on average:
> >
> > [1] Choe et al., “Evaluating Weakly Supervised Object Localization Methods Right,” CVPR 2020.
> >
> > **Table: *Unsupervised ImageNet-1K localization* (MaxBoxAccV2).**
> > | **Model** | **Arch**  | **[CLS]→patch** | **EP** | **Δacc (%)** |
> > |-----------|-----------|-------:|-------:|-------------:|
> > | MAE       | ViT-S/16  |  46.0  |  58.5  |       +12.5  |
> > | MAE       | ViT-B/16  |  54.2  |  60.4  |        +6.2  |
> > | MAE       | ViT-L/16  |  46.9  |  61.2  |       +14.3  |
> > | BEiTv2    | ViT-B/16  |  47.0  |  61.2  |       +14.2  |
> > | SimMIM    | ViT-B/16  |  45.2  |  60.0  |       +14.8  |
> > | iBOT      | ViT-B/16  |  57.6  |  63.7  |        +6.1  |
> > | SigLIP    | ViT-L/16  |  44.0  |  44.2  |        +0.2  |
> > | **Average** |           | **48.7** | **58.5** |     **+9.8** |
> >
> > These results show that EP’s attention maps act as strong unsupervised localizers *out-of-the-box*. We have included these results in revised Appendix subsection C.3 (Table 6).
> >
> > ----
> >
> > ### EP for retrieval
> >
> > Following the standard R@K (Recall@K) protocol, we evaluate EP on two fine-grained datasets, CUB200 and Cars196 (Tables below), using the same EP trained via attentive probing on ImageNet-1K. On CUB, EP consistently boosts performance across five models (e.g., +40.8% for MAE, +11.2% for BEiTv2, +10.3% for iBOT, +5.3% for CLIP, +15.9% for SigLIP on R@1 metric). Similarly on Cars196 the gains remain significant in most cases (e.g., +21.5% for MAE, +8.6% for iBOT, +4.6% for SigLIP). These findings show that EP features can be used *out-of-the-box* for image retrieval and consistently outperform the features of the frozen backbones. We have included these results in revised Appendix subsection C.3 (Tables 7,8).
> >
> > **Table: Zero-shot Image Retrieval performance (R@K) on *CUB200* (images: 11788, classes: 200) with and without EP.**
> > | **Features**                 | R@1  | R@2  | R@4  | R@8  | R@16 | R@32 |
> > |-----------------------------|-----:|-----:|-----:|-----:|-----:|-----:|
> > | frozen MAE backbone         | 16.3 | 23.7 | 33.3 | 45.0 | 58.0 | 70.8 |
> > | + EP on ImageNet            | 57.1 | 69.5 | 79.5 | 87.5 | 93.0 | 96.2 |
> > | frozen BEiTv2 backbone      | 57.0 | 68.2 | 78.0 | 85.8 | 90.9 | 94.2 |
> > | + EP on ImageNet            | 68.2 | 78.7 | 86.7 | 92.3 | 95.9 | 97.8 |
> > | frozen iBOT backbone        | 51.8 | 64.0 | 74.7 | 83.2 | 90.0 | 94.1 |
> > | + EP on ImageNet            | 62.1 | 73.7 | 82.4 | 89.7 | 94.0 | 96.8 |
> > | frozen CLIP backbone        | 75.0 | 84.3 | 91.3 | 95.3 | 97.5 | 98.7 |
> > | + EP on ImageNet            | 80.3 | 88.0 | 92.9 | 95.8 | 97.6 | 98.6 |
> > | frozen SigLIP backbone      | 60.8 | 72.1 | 81.6 | 88.8 | 93.5 | 96.4 |
> > | + EP on ImageNet            | 76.7 | 85.7 | 92.0 | 95.7 | 97.8 | 98.8 |
> >
> > **Table: Zero-shot Image Retrieval performance (R@K) on *Cars196* (images: 16185, classes: 196) with and without EP.**
> > | **Features**                 | R@1  | R@2  | R@4  | R@8  | R@16 | R@32 |
> > |-----------------------------|-----:|-----:|-----:|-----:|-----:|-----:|
> > | frozen MAE backbone         | 12.7 | 17.6 | 23.7 | 31.8 | 42.9 | 55.7 |
> > | + EP on ImageNet            | 34.2 | 44.8 | 55.6 | 65.9 | 76.4 | 85.2 |
> > | frozen BEiTv2 backbone      | 48.3 | 60.7 | 72.3 | 81.8 | 89.2 | 94.5 |
> > | + EP on ImageNet            | 44.2 | 56.4 | 68.5 | 79.1 | 87.3 | 93.7 |
> > | frozen iBOT backbone        | 31.5 | 40.7 | 50.2 | 60.0 | 69.9 | 79.0 |
> > | + EP on ImageNet            | 40.1 | 50.6 | 61.0 | 70.7 | 79.5 | 87.6 |
> > | frozen CLIP backbone        | 79.6 | 88.9 | 94.6 | 97.8 | 99.2 | 99.7 |
> > | + EP on ImageNet            | 79.8 | 89.3 | 94.8 | 97.8 | 99.2 | 99.7 |
> > | frozen SigLIP backbone      | 85.8 | 92.3 | 96.1 | 98.0 | 99.1 | 99.6 |
> > | + EP on ImageNet            | 90.4 | 95.4 | 97.8 | 99.0 | 99.6 | 99.8 |
> >
> > In summary, EP *is not tied to image classification*: both its **attention maps** (queries) and its **pooled features** can be used for non-classification tasks, delivering **strong performance**.
> >
> > ---

---

> > > ### Author Response · Authors · 2025-11-25
> > > **Answer to Reviewer Vb6E (part 3)**
> > >
> > > **EP for large-scale models and datasets.**
> > >
> > > We thank the reviewer for raising this question regarding scalability. As shown in Table 2 of our main paper, our experiments already include **DiT-XL** (**677M parameters**), which is among the largest vision backbones in literature. EP continues to provide substantial accuracy–parameter benefits even at this scale (+24.3\%).
> > >
> > > To further evaluate EP's behavior under very large datasets, following the reviewer’s suggestion, we conduct experiments on **ImageNet-21K** (**14M images and over 19K classes**). Although ImageNet-21K is typically used for pre-training rather than probing, we apply EP using MAE ViT-B for downstream classification. We observe that EP still outperforms linear probing (LP=31.9% vs. EP=35.6%), indicating that EP scales robustly with both model (DiT-XL) and dataset size (ImageNet-21K).
> > >
> > > ---
> > >
> > > **Comparison to zero-shot recognition with VLMs.**
> > >
> > > We thank the reviewer for pointing out the importance of modern low-cost baselines such as zero-shot VLMs.
> > >
> > > To directly address this concern, we evaluate EP on three strong large-scale VLMs (**CLIP ViT-L/14, SigLIP ViT-L/16, SigLIP2 ViT-L/16**) on ImageNet-1K validation set, while keeping the backbone completely frozen. The Table below reports the *zero-shot recognition accuracy* of each VLM together with the performance obtained by t*raining only the EP mechanism on top of the frozen vision encoder*. EP consistently improves over the zero-shot predictions by +7.9, +5.6 and +4.5 points for CLIP, SigLIP and SigLIP2 respectively. This shows that **(i)** strong zero-shot VLMs still benefit significantly from probing, and **(ii)** EP provides a competitive, low-cost alternative for assessing and exploiting pretrained representations without modifying the backbone parameters.
> > >
> > > **Table: Zero-shot ImageNet-1K accuracy of strong VLMs compared to the same backbones probed with Efficient Probing (EP).**  The backbone is kept frozen and only the lightweight EP head is trained.
> > > | Model   | Zero-shot acc. (%) | EP probing acc. (%) | $\Delta$acc. (EP $-$ ZS) |
> > > |---------|--------------------:|--------------------:|--------------------------:|
> > > | CLIP    |               75.5  |               83.4  |                     +7.9  |
> > > | SigLIP  |               80.5  |               86.1  |                     +5.6  |
> > > | SigLIP2 |               82.5  |               87.0  |                     +4.5  |
> > >
> > > ---

---

> ### Author Response · Authors · 2025-11-25
> **Answer to Reviewer Vb6E (part 4)**
>
> **Comparison to PEFT methods.**
>
> We thank the reviewer for pointing out the connection between EP and parameter-efficient fine-tuning (PEFT). Conceptually, we view EP as a form of **parameter-efficient probing** (PEP): the backbone remains frozen and EP is used to evaluate efficiently the quality of pre-trained representations. **Parameter-efficient fine-tuning** (PEFT) techniques such as LoRA constitute a *natural point of comparison*, cause even if the backbone is not frozen, they also train a small number of additional parameters. Although our paper mostly focuses on *probing for evaluation*—rather than adaptation—we agree that contrasting EP with PEFT is highly insightful. This comparison was not included in the original submission *due to space and scope constraints*, but we have now conducted a thorough study and the results provide **valuable insights for the community**.
>
> Concretely, in the r**evised subsection 4.2 and Figure 4** (as also in the Table below) we compare the most efficient probing baselines ([CLS], AbMILP, DELF, AIM), EP at different output dimensions $D_o$, more than 40 LoRA variants (single-layer and all-layer, with various $W_Q, W_K,W_V,W_O$ combinations and ranks $\rho\in\{8,16,32,64\}$), as well as BitFit and LayerNorm tuning, all on a frozen MAE ViT-B/16 backbone. We observe that:
>
> **(i)** **LoRA** applied to a **single** layer (layers 4, 8, or 12; red, cyan, and magenta crosses) lies roughly on or slightly above the baseline probing Pareto front, but **is consistently dominated by EP** in the accuracy–parameter plane. For example, EP$^{D_i/2}$ achieves $74.9\%$ top-1 with $7.5\times 10^5$ parameters, whereas the best single-layer LoRA configuration in that region needs $1.16\times 10^6$ parameters to reach the same performance.
>
> **(ii)** **BitFit** and **LayerNorm tuning** (orange-brown and green crosses) also outperform pure probing baselines, yet **EP remains strictly more parameter-efficient**: EP$^{D_i/2}$ matches or exceeds their accuracy while using fewer parameters.
>
> **(iii)** **All-layer LoRA** configurations (orange crosses) *move closer to the EP Pareto frontier* and some *slightly surpass* EP in accuracy (up to $76.7\%$ with $1.95\times 10^6$ parameters). This is expected, since these configurations perform genuine low-rank fine-tuning of the backbone rather than pure probing.
>
> Motivated by this, we further study the *complementarity* between PEFT and PEP by combining one of the most parameter-efficient LoRA variants (LoRA on all $W_V$ matrices across all layers) with EP at different $D_o$ values (LoRA$+$EP, orange star–cross markers). These hybrids form a new dominant region in the accuracy–parameter plane, strictly improving over both pure EP and pure LoRA. For instance, a LoRA$+$EP configuration with $8.5\times 10^5$ parameters attains $76.99\%$ top-1, outperforming both the best pure EP setting ($75.58\%$ at $1.38\times 10^6$ parameters) and the best all-layer LoRA variant ($76.72\%$ at $1.95\times 10^6$ parameters). At the low-parameter end, a LoRA$+$EP configuration with only $2.5\times 10^5$ parameters already reaches $71.99\%$, i.e., about $4.3\%$ above [CLS] linear probing ($67.66\%$) while using more than $3\times$ fewer parameters ([CLS] uses $7.7\times 10^5$ parameters).

---

> > ### Author Response · Authors · 2025-11-25
> > **Answer to Reviewer Vb6E (part 5)**
> >
> > **Table: Accuracy–parameter trade-off for baseline probing methods, EP variants, LoRA configurations, and hybrid LoRA+EP / BitFit+EP settings on MAE ViT-B/16.**
> >
> > | Method                     | # Params | Acc. (%) | Layer | Applied to                      | $\rho$ | $\alpha$ |
> > |----------------------------|---------:|---------:|:------|:--------------------------------|:------:|:--------:|
> > | CLS                        |  769000  |   67.7   | --    | --                              |  --    |   --     |
> > | AIM                        | 1949416  |   75.12  | --    | --                              |  --    |   --     |
> > | AbMILP                     |  769769  |   71.74  | --    | --                              |  --    |   --     |
> > | LoRA\_Q12              |  781288  |   71.21  | 12    | $W_Q$                           |   8    |   16     |
> > | LoRA\_K12                  |  781288  |   71.33  | 12    | $W_K$                           |   8    |   16     |
> > | LoRA\_V12                  |  781288  |   68.67  | 12    | $W_V$                           |   8    |   16     |
> > | LoRA\_QK12                 |  793576  |   71.79  | 12    | $W_Q{+}W_K$                     |   8    |   16     |
> > | LoRA\_QKV12                |  805864  |   72.71  | 12    | $W_Q{+}W_K{+}W_V$               |   8    |   16     |
> > | LoRA\_QKVO12               |  818152  |   72.75  | 12    | $W_Q{+}W_K{+}W_V{+}W_O$         |   8    |   16     |
> > | LoRA\_QKVO12\_rho16        |  867304  |   73.65  | 12    | $W_Q{+}W_K{+}W_V{+}W_O$         |  16    |   32     |
> > | LoRA\_QKVO12\_rho32        |  965608  |   74.30  | 12    | $W_Q{+}W_K{+}W_V{+}W_O$         |  32    |   64     |
> > | LoRA\_QKVO12\_rho64        | 1162216  |   74.90  | 12    | $W_Q{+}W_K{+}W_V{+}W_O$         |  64    |  128     |
> > | LoRA\_Q4                   |  781288  |   69.11  |  4    | $W_Q$                           |   8    |   16     |
> > | LoRA\_K4                   |  781288  |   69.09  |  4    | $W_K$                           |   8    |   16     |
> > | LoRA\_V4                   |  781288  |   70.99  |  4    | $W_V$                           |   8    |   16     |
> > | LoRA\_QK4                  |  793576  |   69.23  |  4    | $W_Q{+}W_K$                     |   8    |   16     |
> > | LoRA\_QKV4                 |  805864  |   71.48  |  4    | $W_Q{+}W_K{+}W_V$               |   8    |   16     |
> > | LoRA\_QKVO4                |  818152  |   71.94  |  4    | $W_Q{+}W_K{+}W_V{+}W_O$         |   8    |   16     |
> > | LoRA\_QKVO4\_rho16         |  867304  |   71.89  |  4    | $W_Q{+}W_K{+}W_V{+}W_O$         |  16    |   32     |
> > | LoRA\_QKVO4\_rho32         |  965608  |   72.31  |  4    | $W_Q{+}W_K{+}W_V{+}W_O$         |  32    |   64     |
> > | LoRA\_QKVO4\_rho64         | 1162216  |   72.27  |  4    | $W_Q{+}W_K{+}W_V{+}W_O$         |  64    |  128     |
> > | LoRA\_Q8                   |  781288  |   70.00  |  8    | $W_Q$                           |   8    |   16     |
> > | LoRA\_K8                   |  781288  |   69.70  |  8    | $W_K$                           |   8    |   16     |
> > | LoRA\_V8                   |  781288  |   71.89  |  8    | $W_V$                           |   8    |   16     |
> > | LoRA\_QK8                  |  793576  |   70.25  |  8    | $W_Q{+}W_K$                     |   8    |   16     |
> > | LoRA\_QKV8                 |  805864  |   72.83  |  8    | $W_Q{+}W_K{+}W_V$               |   8    |   16     |
> > | LoRA\_QKVO8                |  818152  |   72.84  |  8    | $W_Q{+}W_K{+}W_V{+}W_O$         |   8    |   16     |
> > | LoRA\_QKVO8\_rho16         |  867304  |   73.31  |  8    | $W_Q{+}W_K{+}W_V{+}W_O$         |  16    |   32     |
> > | LoRA\_QKVO8\_rho32         |  965608  |   73.52  |  8    | $W_Q{+}W_K{+}W_V{+}W_O$         |  32    |   64     |
> > | LoRA\_QKVO8\_rho64         | 1162216  |   73.73  |  8    | $W_Q{+}W_K{+}W_V{+}W_O$         |  64    |  128     |

---

> > > ### Author Response · Authors · 2025-11-25
> > > **Answer to Reviewer Vb6E (part 6)**
> > >
> > > **Table: Accuracy–parameter trade-off for baseline probing methods, EP variants, LoRA configurations, and hybrid LoRA+EP / BitFit+EP settings on MAE ViT-B/16.**
> > >
> > > | Method                     | # Params | Acc. (%) | Layer | Applied to                      | $\rho$ | $\alpha$ |
> > > |----------------------------|---------:|---------:|:------|:--------------------------------|:------:|:--------:|
> > > | LoRA\_Qall                 |  916456  |   74.29  | all   | $W_Q$                           |   8    |   16     |
> > > | LoRA\_Kall                 |  916456  |   74.49  | all   | $W_K$                           |   8    |   16     |
> > > | LoRA\_Vall                 |  916456  |   74.89  | all   | $W_V$                           |   8    |   16     |
> > > | LoRA\_QKall                | 1063912  |   74.96  | all   | $W_Q{+}W_K$                     |   8    |   16     |
> > > | LoRA\_QKV\_all             | 1211368  |   76.01  | all   | $W_Q{+}W_K{+}W_V$               |   8    |   16     |
> > > | LoRA\_QKVO\_all            | 1358824  |   76.36  | all   | $W_Q{+}W_K{+}W_V{+}W_O$         |   8    |   16     |
> > > | LoRA\_Qall\_rho16          | 1063912  |   74.62  | all   | $W_Q$                           |  16    |   32     |
> > > | LoRA\_QKall\_rho16         | 1358824  |   75.05  | all   | $W_Q{+}W_K$                     |  16    |   32     |
> > > | LoRA\_QKVall\_rho16        | 1653736  |   76.46  | all   | $W_Q{+}W_K{+}W_V$               |  16    |   32     |
> > > | LoRA\_QKVOall\_rho16       | 1948648  |   76.72  | all   | $W_Q{+}W_K{+}W_V{+}W_O$         |  16    |   32     |
> > > | LoRA\_QKVOall\_rho32       | 3128296  |   77.06  | all   | $W_Q{+}W_K{+}W_V{+}W_O$         |  32    |   64     |
> > > | LoRA\_QKVOall\_rho64       | 5487592  |   77.18  | all   | $W_Q{+}W_K{+}W_V{+}W_O$         |  64    |  128     |
> > > | LayerNorm                  |  845800  |   72.80  | --    | --                              |  --    |   --     |
> > > | BitFit                     |  975824  |   74.71  | --    | --                              |  --    |   --     |
> > > | EP$^{D_i}$                 | 1395688  |   75.58  | --    | --                              |  --    |   --     |
> > > | EP$^{D_i/2}$               |  753640  |   74.86  | --    | --                              |  --    |   --     |
> > > | EP$^{D_i/4}$               |  389608  |   72.88  | --    | --                              |  --    |   --     |
> > > | EP$^{D_i/8}$               |  207592  |   70.27  | --    | --                              |  --    |   --     |
> > > | LoRA\_Vall+EP$^{D_i}$      | 1530856  |   77.23  | all   | $W_V$                           |   8    |   16     |
> > > | LoRA\_QKVall+EP$^{D_i}$    | 1825768  |   77.52  | all   | $W_Q{+}W_K{+}W_V$               |   8    |   16     |
> > > | LoRA\_QKVOall+EP$^{D_i}$   | 1973224  |   77.51  | all   | $W_Q{+}W_K{+}W_V{+}W_O$         |   8    |   16     |
> > > | LoRA\_Vall+EP$^{D_i/2}$    |  851944  |   76.99  | all   | $W_V$                           |   8    |   16     |
> > > | LoRA\_Vall+EP$^{D_i/4}$    |  512488  |   75.84  | all   | $W_V$                           |   8    |   16     |
> > > | LoRA\_Vall+EP$^{D_i/8}$    |  342760  |   74.17  | all   | $W_V$                           |   8    |   16     |
> > > | LoRA\_Vall+EP$^{D_i/16}$   |  251752  |   71.99  | all   | $W_V$                           |   8    |   16     |
> > > | BitFit+EP$^{D_i}$          | 1590224  |   77.28  | --    | --                              |  --    |   --     |
> > > | BitFit+EP$^{D_i/2}$        |  911312  |   77.02  | --    | --                              |  --    |   --     |
> > > | BitFit+EP$^{D_i/4}$        |  571856  |   75.41  | --    | --                              |  --    |   --     |
> > > | BitFit+EP$^{D_i/8}$        |  402128  |   73.54  | --    | --                              |  --    |   --     |

---

> ### Author Response · Authors · 2025-11-25
> **Answer to Reviewer Vb6E (part 7)**
>
> To better understand how EP and LoRA modify the representation, we also perform a cross-dataset $k$-NN evaluation on MAE ViT-B features (revised Appendix Table 5 - Table below) using ImageNet-1K, Cars196, and Food101. For each dataset, we compare: (a) the frozen MAE backbone, (b) features derived via EP (EP$_{32}$) trained on that dataset, and (c) features from the backbone fine-tuned with the best-performing LoRA configuration. As expected, LoRA achieves the strongest in-domain performance (e.g., $72.3\%$ on ImageNet-1K, $75.4\%$ on Cars196, $80.3\%$ on Food101), reflecting its role as an adaptation mechanism. However, EP consistently provides stronger or comparable out-of-domain performance: for example, when trained on Cars196, EP features yield $51.7\%$ on ImageNet-1K and $42.0\%$ on Food101, versus $45.8\%$ and $38.7\%$ for LoRA; when trained on Food101, EP achieves $58.2\%$ on ImageNet-1K and $23.4\%$ on Cars196, compared to $56.7\%$ and $15.2\%$ for LoRA.
>
> **Table: *Cross-dataset k-NN evaluation on MAE ViT-B using frozen, EP-probed, and LoRA-tuned features*.**
> Default EP corresponds to EP$_{32}$, while the “best LoRA” configuration applies LoRA to all 12 layers on the $W_Q,Q_K,W_V$, and $W_O$ projection matrices with rank $\rho{=}8$.
>
> | **Features**                          | **ImageNet-1K** | **Cars196** | **Food101** |
> |--------------------------------------|----------------:|-----------------:|------------:|
> | frozen MAE backbone                  |            46.1 |              9.5 |        28.8 |
> | + EP probed on ImageNet-1K           |            70.5 |           **31.4** |        64.2 |
> | + EP probed on Cars196          |          **51.7** |             70.0 |      **42.0** |
> | + EP probed on Food-101              |          **58.2** |           **23.4** |        75.2 |
> | + best LoRA tuned on ImageNet-1K     |          **72.3** |             24.7 |      **65.6** |
> | + best LoRA tuned on Cars196    |            45.8 |           **75.4** |        38.7 |
> | + best LoRA tuned on Food-101        |            56.7 |             15.2 |      **80.3** |
>
> Taken together, these results: (1) show that EP is competitive with, and often strictly more parameter-efficient than, standard PEFT baselines; and (2) they highlight that LoRA and EP operate in *complementary* regimes—LoRA specializes the backbone to a specific task, while EP preserves and probes the pre-trained representation—so that combining them (LoRA$+$EP) yields the best accuracy–efficiency trade-offs. Overall, our findings indicate that incorporating EP is consistently beneficial, whether used on its own or alongside PEFT methods.
>
> ----

---

> > ### Author Response · Authors · 2025-11-25
> > **Answer to Reviewer Vb6E (part 8)**
> >
> > **Computational cost, 90-epoch schedule, frozen-feature evaluation.**
> >
> > We thank the reviewer for raising this point. Our goal in this work is to compare different probing protocols under a **common, standardized** evaluation setting. For this reason, we follow prior work and train *all* probes (LP, AIM, AbMILP, V-JEPA, EP) for 90 epochs with **identical, mild data augmentations**. This ensures that differences in performance are attributable to the probing mechanism rather than to training schedules.
> >
> > ### Convergence in practice
> >
> > Although we use 90 epochs for fairness, EP converges **very quickly** in practice. Below we report top-1 ImageNet-1K validation accuracy at *epoch 10* and *epoch 90* for 12 representative backbones:
> >
> > **Table: *EP convergence on ImageNet-1K.* EP@10 is EP accuracy after 10 epochs; LP@90 and EP@90 are final probing results after 90 epochs.**
> >
> > | Model              | EP@10 | LP@90 | EP@90 | EP@10 vs. LP@90 Gap (%) | EP@10 vs. EP@90 Gap (%) |
> > |--------------------|:-----:|:-----:|:-----:|:------------------------:|:------------------------:|
> > | MAE ViT-B/16       | 60.0  | 67.7  | 75.6  | 88.6                     | 79.4                     |
> > | BEiTv2 ViT-B/16    | 80.0  | 79.0  | 81.7  | 101.3                    | 97.9                     |
> > | CAPI ViT-L/14      | 82.3  | 81.5  | 83.6  | 101.0                    | 98.4                     |
> > | iBOT ViT-B/16      | 78.6  | 78.7  | 79.2  | 99.9                     | 99.2                     |
> > | DINOv2 ViT-B/14    | 83.6  | 83.2  | 84.0  | 100.5                    | 99.5                     |
> > | DINOv2 ViT-L/14    | 85.0  | 85.2  | 85.6  | 99.8                     | 99.3                     |
> > | Franca ViT-B/16    | 83.8  | 83.8  | 84.3  | 100.0                    | 99.4                     |
> > | DINOv3 ViT-B/16    | 84.4  | 84.0  | 84.4  | 100.5                    | 100.0                    |
> > | DINOv3 ViT-L/14    | 86.7  | 86.6  | 87.1  | 100.1                    | 99.5                     |
> > | CLIP ViT-L/14      | 82.7  | 82.3  | 83.4  | 100.5                    | 99.2                     |
> > | SigLIP ViT-L/16    | 85.6  | 84.1  | 86.1  | 101.8                    | 99.4                     |
> > | AIMv2 ViT-L/14     | 85.5  | 84.8  | 85.9  | 100.8                    | 99.5                     |
> >
> >
> > Across these models, after 10 training epochs, EP already matches, and in most cases surpasses, standard linear probing (LP) performance trained for the full 90 epochs. This is reflected in the EP@10 vs. LP@90 column of the table above, where the corresponding gap is nearly 100% or more. Similarly, EP's own convergence is remarkably fast, recovering over 97% of its final performance on average, across all models reported. In the revised version, we have included this in Appendix subsection C.3 Table 9.
> >
> > ### Wall-clock cost
> >
> > On an NVIDIA RTX A5000, the wall-clock difference between LP and EP is very small: full-dimensional EP ($D_o = D_i$) is only 20 seconds slower per epoch than LP, while eighth-dimensional EP ($D_o = D_i/8$) is actually 15 seconds *faster* than LP. Moreover, EP can operate at lower output dimensionality with better accuracy than LP. For example, for MAE ViT-B/16 on ImageNet-1K, EP at $D_o = D_i/8$ achieves 70.3% top-1 accuracy with only 207,592 parameters, whereas LP at $D$ reaches 67.7% using almost $4\times$ more parameters (already noted in L322–323 of the main paper).
> >
> > Given these, we *respectfully disagree* with the claim that the computational cost of EP is “much higher” than that of standard LP.
> >
> > ### Frozen-feature evaluation
> >
> > We agree that probing on frozen and pre-stored features is a promising direction, and we *explicitly mention this as future work*. However, this setting introduces additional challenges: (i) all patch-token features must be stored on disk, which is considerably heavier than storing a single CLS token per image; and (ii) input-level augmentations can no longer be applied, so one must rely on feature-space augmentations, which are less well understood.
> >
> > To quantify the effect, we carry out an additional experiment on MAE ViT-B/16 with frozen features on ImageNet-1K. In this regime, LP reaches 65.9% and EP reaches 73.6%, i.e. both incur a drop of roughly 2 points compared to our on-the-fly current setting. We believe that this gap can be reduced by designing better feature-level augmentations (e.g. in the spirit of [2]), but fully exploring this direction is beyond the scope of the present work. We have revised section D of Appendix adding a paragraph about this.
> >
> > [2] Bär et al., “Frozen Feature Augmentation for Few-Shot Image Classification,” CVPR 2024.

---

> > > ### Author Response · Authors · 2025-11-25
> > > **Answer to Reviewer Vb6E (part 9 - last part)**
> > >
> > > **Motivation for removing $W_K$ and relation to MHCA.**
> > >
> > > We appreciate the reviewer’s request for a clearer justification. Our claim is that for the same number of queries/heads, EP is a more *parameter- and compute-efficient re-parameterization*. As discussed in subsection 3.2, MHCA, that natively includes $W_K$, is mathematically equivalent to MQCA, that natively does not include $W_K$, which is also confirmed empirically by the identical performance of AIM with 12 queries (MHCA) and EP with 12 queries (MQCA), both at 75.1\% (revised subsection 4.3).
> > >
> > > The poor performance of the partially shared–subspace variant in Eq. (8) is therefore not evidence that “projected” queries are inherently worse, but that restricting each head to a subspace of the feature dimension reduces effective capacity. We have revised the text around Eqs. (7–10) to highlight this distinction and to emphasize that EP’s advantage comes from removing redundant projections in MHCA (thus avoiding over-parameterization in the probing setting), while keeping the value projection $W_V$, whose importance we ablate in the main paper.
> > >
> > > -----
> > >
> > > **Hypothetical future efficient method.**
> > > We thank the reviewer for the comment. Attentive probing is indeed a relatively new evaluation protocol, which is precisely why existing methods have not been optimized for parameter– or compute–efficiency. Our intention is not to claim that prior work is “weak,” but that EP addresses **concrete design limitations that exist in the literature** (redundant projections, large MLP blocks, etc.).
> > >
> > > To ensure fairness, we evaluated strong representative baselines (AIM, AbMILP, SimPool, V-JEPA) and additionally, during this rebuttal, we **additionally included a broad suite of PEFT methods** (many LoRA variants, BitFit, tuning LayerNorm). These results show that EP **remains competitive** even when compared to highly optimized parameter-efficient tuning strategies. Moreover, EP is *complementary* to PEFT: **combining EP with LoRA yields hybrid configurations that strictly improve the accuracy–parameter trade-off over using either EP or LoRA alone**.
> > >
> > > We agree that future methods may further improve this space. This is precisely why we developed EP: to provide a simple, efficient, and well-understood baseline upon which others can build. We would *genuinely welcome stronger future approaches*.
> > >
> > > Our work aims to make this progress easier and clearer.

---

> > > > ### Comment · Reviewer_Vb6E · 2025-11-26
> > > >
> > > > Thank you for the authors’ response, which has addressed some of my concerns. As I mentioned in Question 3, in the era of large models, the training of visual backbones may ultimately use large models as encoders. How to quickly and cost-effectively reflect the true capabilities of a backbone during training is of greater industrial value. Since EP requires additional training, this remains my main concern.
> > > >
> > > > I have decided to raise my score from 2 to 4. Whether the paper is accepted as an ICLR conference paper will be comprehensively evaluated during the discussion phase among the reviewers and the Area Chair.

---

> > > > > ### Author Response · Authors · 2025-12-03
> > > > > **Answer to Reviewer Vb6E**
> > > > >
> > > > > We would like to thank the reviewer for raising their score to 4 and for acknowledging that some of their concerns have been addressed.
> > > > >
> > > > > To further speak to the issue of quickly reflecting backbone quality “in the era of large models”, we additionally experimented with integrating EP directly into the pre-training of a CLIP model on YFCC-100M. In this setting, EP is trained *jointly* with the backbone and text encoder, and at downstream time we perform zero-shot ImageNet-1K recognition *without any additional training*: we simply replace the standard [CLS] with EP's pooled vector. This yields *36.7\%* zero-shot top-1 accuracy with CLIP+EP, compared to *33.4\%* for the standard CLIP vision head, showing that EP can also act as a built-in, cost-free probe.

---

### Official Review · Reviewer_Ph7P · 2025-10-30

**Soundness:** 3
**Presentation:** 3
**Contribution:** 3
**Rating:** 6
**Confidence:** 4

**Summary:**

This paper takes a hard look at how we evaluate giant, pre-trained vision models. The core idea is that full fine-tuning is just too expensive , but the standard alternative, linear probing (LP), doesn't do justice to models (like MIM) that spread information across patch tokens instead of a single [CLS] token. The paper champions "attentive probing"—using an attention head to smartly aggregate patch features—as the right middle ground.

**Strengths:**

1. The biggest strength, in my view, is the systematic benchmark. The field of attentive probing has been a bit all over the place, with different papers (AIM, V-JEPA, etc.) all proposing their own one-off solutions .
2. EP is a nice piece of engineering. It's not flashy, but it's simple, well-motivated (why have redundant projections?), and it just plain works. It consistently lands on the Pareto frontier for both parameter count and GFLOPs, which is exactly what you want from something called "efficient."
3. The experimental setup is comprehensive. The authors didn't just test on one model; they covered the gamut of pre-training paradigms (MIM, JEA, VLMs, generative) across a ton of datasets. The extra analyses, like the low-shot probing (Table 3) and layer-wise probing (Fig. 10), really strengthen the case that EP is a robust tool.

**Weaknesses:**

1. While EP is effective, its novelty is a bit thin. The core idea is essentially an ablation study on Multi-Head Cross-Attention (MHCA),
2. The paper frames this entire problem as "probing for evaluation." But what they're doing—freezing the backbone and training a few extra parameters—is exactly what Parameter-Efficient Fine-Tuning (PEFT) is. The authors even acknowledge EP fits into the PEFT family in Appendix A.2. So, why is there no comparison to mainstream PEFT methods like LoRA, Adapters, or even simpler baselines like BitFit (tuning biases) or just tuning the LayerNorms?
3. The analysis showing that EP's queries are "complementary" is interesting. But the paper never really closes the loop. Is this complementarity the reason EP performs well, or just a neat side effect? Do other methods (like AIM, which Figure 6 shows is also pretty complementary) also benefit? It's presented as a key finding but feels disconnected from the main efficiency argument.

**Questions:**

1. The main question is about the PEFT comparisons. Could you provide data on how EP stacks up against training a LoRA module (with a frozen backbone) using an equivalent number of trainable parameters?
2. Regarding the complementarity analysis: do you have any experiments to suggest this is a causal driver of performance? For example, what happens if you explicitly add a loss term to force the queries in AIM or V-JEPA to be more complementary? Would their performance increase?

---

> ### Author Response · Authors · 2025-11-25
> **Answer to Reviewer Ph7P (part 1)**
>
> We thank **Reviewer Ph7P** for their *thoughtful* and *insightful* review, and for the **clear recognition of the motivation, scope, and contributions** of our work. We are grateful for the reviewer’s appreciation of the **systematic benchmarking effort**, the **simplicity** and **effectiveness** of EP, and the **breadth of our evaluation** across diverse pre-training paradigms and datasets. We also appreciate the reviewer’s acknowledgement of the **strength of our additional analyses**, such as low-shot and layer-wise probing. Importantly, the reviewer’s questions regarding the relationship between LoRA and attentive probing *prompted us to extend our analysis to a broader comparison between PEFT and probing*—an angle we believe provides **meaningful value to the community**. We address the reviewer’s comments in detail below.
>
> -----
>
> **PEFT-EP comparison and analysis.**
>
> We thank the reviewer for pointing out the connection between EP and parameter-efficient fine-tuning (PEFT). Conceptually, we view EP as a form of **parameter-efficient probing** (PEP): the backbone remains frozen and EP is used to evaluate efficiently the quality of pre-trained representations. **Parameter-efficient fine-tuning** (PEFT) techniques such as LoRA constitute a *natural point of comparison*, cause even if the backbone is not frozen (e.g., in BitFit), they also train a small number of additional parameters. Although our paper mostly focuses on *probing for evaluation*—rather than adaptation—we agree that contrasting EP with PEFT is highly insightful. This comparison was not included in the original submission *due to space and scope constraints*, but we have now conducted a thorough study and the results provide **valuable insights for the community**.
>
> Concretely, in the revised subsection 4.2 and Figure 4 (as also in the Table below) we compare the most efficient probing baselines ([CLS], AbMILP, DELF, AIM), EP at different output dimensions $D_o$, more than 40 LoRA variants (single-layer and all-layer, with various $W_Q, W_K,W_V,W_O$ combinations and ranks $\rho\in\{8,16,32,64\}$), as well as BitFit and LayerNorm tuning, all on a frozen MAE ViT-B/16 backbone. We observe that:
>
> **(i)** **LoRA** applied to a **single** layer (layers 4, 8, or 12; red, cyan, and magenta crosses) lies roughly on or slightly above the baseline probing Pareto front, but **is consistently dominated by EP** in the accuracy–parameter plane. For example, EP$^{D_i/2}$ achieves $74.9\%$ top-1 with $7.5\times 10^5$ parameters, whereas the best single-layer LoRA configuration in that region needs $1.16\times 10^6$ parameters to reach the same performance.
>
> **(ii)** **BitFit** and **LayerNorm tuning** (orange-brown and green crosses) also outperform pure probing baselines, yet **EP remains strictly more parameter-efficient**: EP$^{D_i/2}$ matches or exceeds their accuracy while using fewer parameters.
>
> **(iii)** **All-layer LoRA** configurations (orange crosses) *move closer to the EP Pareto frontier* and some *slightly surpass* EP in accuracy (up to $76.7\%$ with $1.95\times 10^6$ parameters). This is expected, since these configurations perform genuine low-rank fine-tuning of the backbone rather than pure probing.
>
> Motivated by this, we further study the *complementarity* between PEFT and PEP by combining one of the most parameter-efficient LoRA variants (LoRA on all $W_V$ matrices across all layers) with EP at different $D_o$ values (LoRA$+$EP, orange star–cross markers). These hybrids form a new dominant region in the accuracy–parameter plane, strictly improving over both pure EP and pure LoRA. For instance, a LoRA$+$EP configuration with $8.5\times 10^5$ parameters attains $76.99\%$ top-1, outperforming both the best pure EP setting ($75.58\%$ at $1.38\times 10^6$ parameters) and the best all-layer LoRA variant ($76.72\%$ at $1.95\times 10^6$ parameters). At the low-parameter end, a LoRA$+$EP configuration with only $2.5\times 10^5$ parameters already reaches $71.99\%$, i.e., about $4.3\%$ above [CLS] linear probing ($67.66\%$) while using more than $3\times$ fewer parameters ([CLS] uses $7.7\times 10^5$ parameters).

---

> ### Author Response · Authors · 2025-11-25
> **Answer to Reviewer Ph7P (part 2)**
>
> **Table: Accuracy–parameter trade-off for baseline probing methods, EP variants, LoRA configurations, and hybrid LoRA+EP / BitFit+EP settings on MAE ViT-B/16.**
>
> | Method                     | # Params | Acc. (%) | Layer | Applied to                      | $\rho$ | $\alpha$ |
> |----------------------------|---------:|---------:|:------|:--------------------------------|:------:|:--------:|
> | CLS                        |  769000  |   67.7   | --    | --                              |  --    |   --     |
> | AIM                        | 1949416  |   75.12  | --    | --                              |  --    |   --     |
> | AbMILP                     |  769769  |   71.74  | --    | --                              |  --    |   --     |
> | LoRA\_Q12              |  781288  |   71.21  | 12    | $W_Q$                           |   8    |   16     |
> | LoRA\_K12                  |  781288  |   71.33  | 12    | $W_K$                           |   8    |   16     |
> | LoRA\_V12                  |  781288  |   68.67  | 12    | $W_V$                           |   8    |   16     |
> | LoRA\_QK12                 |  793576  |   71.79  | 12    | $W_Q{+}W_K$                     |   8    |   16     |
> | LoRA\_QKV12                |  805864  |   72.71  | 12    | $W_Q{+}W_K{+}W_V$               |   8    |   16     |
> | LoRA\_QKVO12               |  818152  |   72.75  | 12    | $W_Q{+}W_K{+}W_V{+}W_O$         |   8    |   16     |
> | LoRA\_QKVO12\_rho16        |  867304  |   73.65  | 12    | $W_Q{+}W_K{+}W_V{+}W_O$         |  16    |   32     |
> | LoRA\_QKVO12\_rho32        |  965608  |   74.30  | 12    | $W_Q{+}W_K{+}W_V{+}W_O$         |  32    |   64     |
> | LoRA\_QKVO12\_rho64        | 1162216  |   74.90  | 12    | $W_Q{+}W_K{+}W_V{+}W_O$         |  64    |  128     |
> | LoRA\_Q4                   |  781288  |   69.11  |  4    | $W_Q$                           |   8    |   16     |
> | LoRA\_K4                   |  781288  |   69.09  |  4    | $W_K$                           |   8    |   16     |
> | LoRA\_V4                   |  781288  |   70.99  |  4    | $W_V$                           |   8    |   16     |
> | LoRA\_QK4                  |  793576  |   69.23  |  4    | $W_Q{+}W_K$                     |   8    |   16     |
> | LoRA\_QKV4                 |  805864  |   71.48  |  4    | $W_Q{+}W_K{+}W_V$               |   8    |   16     |
> | LoRA\_QKVO4                |  818152  |   71.94  |  4    | $W_Q{+}W_K{+}W_V{+}W_O$         |   8    |   16     |
> | LoRA\_QKVO4\_rho16         |  867304  |   71.89  |  4    | $W_Q{+}W_K{+}W_V{+}W_O$         |  16    |   32     |
> | LoRA\_QKVO4\_rho32         |  965608  |   72.31  |  4    | $W_Q{+}W_K{+}W_V{+}W_O$         |  32    |   64     |
> | LoRA\_QKVO4\_rho64         | 1162216  |   72.27  |  4    | $W_Q{+}W_K{+}W_V{+}W_O$         |  64    |  128     |
> | LoRA\_Q8                   |  781288  |   70.00  |  8    | $W_Q$                           |   8    |   16     |
> | LoRA\_K8                   |  781288  |   69.70  |  8    | $W_K$                           |   8    |   16     |
> | LoRA\_V8                   |  781288  |   71.89  |  8    | $W_V$                           |   8    |   16     |
> | LoRA\_QK8                  |  793576  |   70.25  |  8    | $W_Q{+}W_K$                     |   8    |   16     |
> | LoRA\_QKV8                 |  805864  |   72.83  |  8    | $W_Q{+}W_K{+}W_V$               |   8    |   16     |
> | LoRA\_QKVO8                |  818152  |   72.84  |  8    | $W_Q{+}W_K{+}W_V{+}W_O$         |   8    |   16     |
> | LoRA\_QKVO8\_rho16         |  867304  |   73.31  |  8    | $W_Q{+}W_K{+}W_V{+}W_O$         |  16    |   32     |
> | LoRA\_QKVO8\_rho32         |  965608  |   73.52  |  8    | $W_Q{+}W_K{+}W_V{+}W_O$         |  32    |   64     |
> | LoRA\_QKVO8\_rho64         | 1162216  |   73.73  |  8    | $W_Q{+}W_K{+}W_V{+}W_O$         |  64    |  128     |

---

> > ### Author Response · Authors · 2025-11-25
> > **Answer to Reviewer Ph7P (part 3)**
> >
> > **Table: Accuracy–parameter trade-off for baseline probing methods, EP variants, LoRA configurations, and hybrid LoRA+EP / BitFit+EP settings on MAE ViT-B/16.**
> >
> > | Method                     | # Params | Acc. (%) | Layer | Applied to                      | $\rho$ | $\alpha$ |
> > |----------------------------|---------:|---------:|:------|:--------------------------------|:------:|:--------:|
> > | LoRA\_Qall                 |  916456  |   74.29  | all   | $W_Q$                           |   8    |   16     |
> > | LoRA\_Kall                 |  916456  |   74.49  | all   | $W_K$                           |   8    |   16     |
> > | LoRA\_Vall                 |  916456  |   74.89  | all   | $W_V$                           |   8    |   16     |
> > | LoRA\_QKall                | 1063912  |   74.96  | all   | $W_Q{+}W_K$                     |   8    |   16     |
> > | LoRA\_QKV\_all             | 1211368  |   76.01  | all   | $W_Q{+}W_K{+}W_V$               |   8    |   16     |
> > | LoRA\_QKVO\_all            | 1358824  |   76.36  | all   | $W_Q{+}W_K{+}W_V{+}W_O$         |   8    |   16     |
> > | LoRA\_Qall\_rho16          | 1063912  |   74.62  | all   | $W_Q$                           |  16    |   32     |
> > | LoRA\_QKall\_rho16         | 1358824  |   75.05  | all   | $W_Q{+}W_K$                     |  16    |   32     |
> > | LoRA\_QKVall\_rho16        | 1653736  |   76.46  | all   | $W_Q{+}W_K{+}W_V$               |  16    |   32     |
> > | LoRA\_QKVOall\_rho16       | 1948648  |   76.72  | all   | $W_Q{+}W_K{+}W_V{+}W_O$         |  16    |   32     |
> > | LoRA\_QKVOall\_rho32       | 3128296  |   77.06  | all   | $W_Q{+}W_K{+}W_V{+}W_O$         |  32    |   64     |
> > | LoRA\_QKVOall\_rho64       | 5487592  |   77.18  | all   | $W_Q{+}W_K{+}W_V{+}W_O$         |  64    |  128     |
> > | LayerNorm                  |  845800  |   72.80  | --    | --                              |  --    |   --     |
> > | BitFit                     |  975824  |   74.71  | --    | --                              |  --    |   --     |
> > | EP$^{D_i}$                 | 1395688  |   75.58  | --    | --                              |  --    |   --     |
> > | EP$^{D_i/2}$               |  753640  |   74.86  | --    | --                              |  --    |   --     |
> > | EP$^{D_i/4}$               |  389608  |   72.88  | --    | --                              |  --    |   --     |
> > | EP$^{D_i/8}$               |  207592  |   70.27  | --    | --                              |  --    |   --     |
> > | LoRA\_Vall+EP$^{D_i}$      | 1530856  |   77.23  | all   | $W_V$                           |   8    |   16     |
> > | LoRA\_QKVall+EP$^{D_i}$    | 1825768  |   77.52  | all   | $W_Q{+}W_K{+}W_V$               |   8    |   16     |
> > | LoRA\_QKVOall+EP$^{D_i}$   | 1973224  |   77.51  | all   | $W_Q{+}W_K{+}W_V{+}W_O$         |   8    |   16     |
> > | LoRA\_Vall+EP$^{D_i/2}$    |  851944  |   76.99  | all   | $W_V$                           |   8    |   16     |
> > | LoRA\_Vall+EP$^{D_i/4}$    |  512488  |   75.84  | all   | $W_V$                           |   8    |   16     |
> > | LoRA\_Vall+EP$^{D_i/8}$    |  342760  |   74.17  | all   | $W_V$                           |   8    |   16     |
> > | LoRA\_Vall+EP$^{D_i/16}$   |  251752  |   71.99  | all   | $W_V$                           |   8    |   16     |
> > | BitFit+EP$^{D_i}$          | 1590224  |   77.28  | --    | --                              |  --    |   --     |
> > | BitFit+EP$^{D_i/2}$        |  911312  |   77.02  | --    | --                              |  --    |   --     |
> > | BitFit+EP$^{D_i/4}$        |  571856  |   75.41  | --    | --                              |  --    |   --     |
> > | BitFit+EP$^{D_i/8}$        |  402128  |   73.54  | --    | --                              |  --    |   --     |

---

> ### Author Response · Authors · 2025-11-25
> **Answer to Reviewer Ph7P (part 4)**
>
> To better understand how EP and LoRA modify the representation, we also perform a cross-dataset $k$-NN evaluation on MAE ViT-B features (revised Appendix Table 5 - Table below) using ImageNet-1K, Cars196, and Food101. For each dataset, we compare: (a) the frozen MAE backbone, (b) features derived via EP (EP$_{32}$) trained on that dataset, and (c) features from the backbone fine-tuned with the best-performing LoRA configuration. As expected, LoRA achieves the strongest in-domain performance (e.g., $72.3\%$ on ImageNet-1K, $75.4\%$ on Cars196, $80.3\%$ on Food101), reflecting its role as an adaptation mechanism. However, EP consistently provides stronger or comparable out-of-domain performance: for example, when trained on Cars196, EP features yield $51.7\%$ on ImageNet-1K and $42.0\%$ on Food101, versus $45.8\%$ and $38.7\%$ for LoRA; when trained on Food101, EP achieves $58.2\%$ on ImageNet-1K and $23.4\%$ on Cars196, compared to $56.7\%$ and $15.2\%$ for LoRA.
>
> **Table: *Cross-dataset k-NN evaluation on MAE ViT-B using frozen, EP-probed, and LoRA-tuned features*.**
> Default EP corresponds to EP$_{32}$, while the “best LoRA” configuration applies LoRA to all 12 layers on the $W_Q,Q_K,W_V$, and $W_O$ projection matrices with rank $\rho{=}8$.
>
> | **Features**                          | **ImageNet-1K** | **Cars196** | **Food101** |
> |--------------------------------------|----------------:|-----------------:|------------:|
> | frozen MAE backbone                  |            46.1 |              9.5 |        28.8 |
> | + EP probed on ImageNet-1K           |            70.5 |           **31.4** |        64.2 |
> | + EP probed on Cars196          |          **51.7** |             70.0 |      **42.0** |
> | + EP probed on Food-101              |          **58.2** |           **23.4** |        75.2 |
> | + best LoRA tuned on ImageNet-1K     |          **72.3** |             24.7 |      **65.6** |
> | + best LoRA tuned on Cars196    |            45.8 |           **75.4** |        38.7 |
> | + best LoRA tuned on Food-101        |            56.7 |             15.2 |      **80.3** |
>
> Taken together, these results: (1) show that EP is competitive with, and often strictly more parameter-efficient than, standard PEFT baselines; and (2) they highlight that LoRA and EP operate in *complementary* regimes—LoRA specializes the backbone to a specific task, while EP preserves and probes the pre-trained representation—so that combining them (LoRA$+$EP) yields the best accuracy–efficiency trade-offs. Overall, our findings indicate that **incorporating EP is consistently beneficial**, whether used on its own or alongside PEFT methods.

---

> ### Author Response · Authors · 2025-11-25
> **Answer to Reviewer Ph7P (part 5 - last part)**
>
> **On the role of complementarity.**
>
> We thank the reviewer for raising this point. In our work, the complementarity metric is primarily used as an *explanatory* and *diagnostic* tool: it quantifies what we visually observe in the attention maps of attentive probes (e.g. Figure 6 for EP). As shown in Figure 7 and detailed in Figure 12 (Appendix), attentive probing heads (AIM, V-JEPA, EP) are consistently more complementary than the internal MHSA heads across backbones and layers, with EP and AIM achieving the highest scores.
>
> To probe causality more directly, we ran an additional experiment where we encourage complementary queries/heads via an auxiliary loss. For AIM, V-JEPA, and EP we add to the cross-entropy loss an extra term $\mathcal{L}_{\text{attn}}$ that penalizes similarity between the attention maps of different queries/heads. On MAE ViT-B probed on ImageNet-1K, this yields a small gain for V-JEPA (74.1\% to 74.3\%), while AIM and EP retain essentially the same final accuracy but **converge faster** (e.g. EP reaches $64.1\%$ at epoch 10 with the loss vs. $60.0\%$ without it, before both variants converge to **the same final score**).
>
> These results suggest that complementarity is indeed a useful property that can modestly help or accelerate training, but for methods such as AIM and EP, which already learn highly complementary queries, the attention mechanism largely discovers this structure on its own. We have revised the Appendix to include a dedicated paragraph on this attention-similarity loss in subsection C.3, and we now provide a plot of validation accuracy over training epochs for EP with and without this loss (Figure 13).
>
> We thank the reviewer once more for their *thoughtful* and *insightful* review.

---

> > ### Comment · Reviewer_Ph7P · 2025-11-26
> >
> > I appreciate the authors' detailed response. The clarifications and the additional context provided have effectively resolved my initial doubts.
> >
> > That said, after re-evaluating the paper and taking into account the points raised by the other reviewers (especially the concerns proposed by Reviewer Vb6E) , I am inclined to maintain my original score of 6. I still view this as a positive rating and I am leaning towards acceptance.

---

> > > ### Author Response · Authors · 2025-12-03
> > > **Answer to Reviewer Ph7P**
> > >
> > > We sincerely thank the reviewer for their thoughtful re-evaluation and for maintaining a positive score leaning toward acceptance. We appreciate that the additional context and clarifications helped resolve the initial doubts.
> > >
> > > Regarding the concerns raised by Reviewer Vb6E, we have now additionally experimented with integrating EP directly into the pre-training (training from scratch) of CLIP model on YFCC-100M. In this setting, EP is trained *jointly* with the backbone and text encoder, and at downstream time we perform zero-shot ImageNet-1K recognition *without any additional training*: we simply replace the standard [CLS] with EP's pooled vector. This yields **36.7\%** zero-shot top-1 accuracy with CLIP+EP, compared to **33.4\%** for the standard CLIP. This result shows that EP can also be used as a ``built-in'' probe that incurs **no extra downstream training cost**.

---

### Official Review · Reviewer_4JLQ · 2025-10-31

**Soundness:** 2
**Presentation:** 3
**Contribution:** 3
**Rating:** 6
**Confidence:** 3

**Summary:**

This paper addresses the challenge of efficiently evaluating large-scale, pre-trained vision models. The authors argue that as "fine-tuning becomes increasingly impractical at scale", probing is emerging as the preferred evaluation protocol. However, standard linear probing (LP) "fails to adequately reflect the potential of models" like Masked Image Models (MIM) or autoregressive models, where information is distributed across patch tokens rather than a single global [CLS] token. This motivates "attentive probing", which uses attention to aggregate patch-level features. The paper notes that existing attentive probing methods suffer from "excessive parameterization and poor computational efficiency". The aothors present the "first comprehensive study of existing methods"  for attentive probing, benchmarking their accuracy and efficiency. They also propose "efficient probing (EP)" , a "simple yet effective multi-query cross-attention mechanism" that "eliminates redundant projections"  to reduce parameter count and computational cost.

**Strengths:**

1. The proposed method, EP, is a "simple multi-query cross-attention mechanism" that is derived by "eliminat[ing] redundant projections"  from standard MHCA. This simplification is well-illustrated in Figure 1.

2. This work provides the "first comprehensive study"  of attentive probing methods. The evaluation is extensive, covering "diverse pre-training paradigms" (MIM, JEA, VLMs, generative) and multiple datasets.

3. EP consistently achieves a state-of-the-art "accuracy-parameter trade-off" , as shown in Figure 2. It achieves "state-of-the-art top-1 accuracy of 75.6% with less than 1.4M parameters" on MAE ViT-B, outperforming more complex methods. The gains are largest for the target model classes, such as "SimMIM +13.6%" and "DiT +24.3%" over LP.

**Weaknesses:**

1. **Placement of Key Ablations**: The justification for EP's specific design is partially buried in Section 4.3. For instance, EP's design is "transformation-free", notably lacking the $W_K$ projection. The empirical justification for this comes from an ablation showing that while removing $W_K$ from multi-head AIM hurts performance ($75.1\% \rightarrow 72.9\%$), EP's design performs well without it. This key comparison, which validates EP's specific design choice, should be more central to the method's motivation in Section 3.

2. **Confusing Terminology**: The method is named "Transformation-free Multi-Query Cross-Attention" , and the text states "there are no projection matrices". However, the value projection $W_V$ is still a critical component of the pooling operation (Equation 2) and the EP method. An ablation in 4.3 confirms that "removing $W_V$ from $EP_{12}$ degrades performance from $75.1\% \rightarrow 72.1\%$". The "transformation-free" label applies only to the key/query interaction, not the value projection, which could be made clearer.

3. **Matryoshka Probing**: The Matryoshka analysis (Appendix C.3, Table 5) is very compelling. It shows that standard EP, when evaluated at smaller dimensions, "collapses" (e.g., 5.5% at $D/8$ ). In contrast, "Efficient Matryoshka" resolves this and provides robust "multi-scale probing without extra parameters". Given this, should "Efficient Matryoshka EP" be the recommended protocol for the community, rather than the standard EP described in the main paper?

**Questions:**

1. **Method Justification (EP vs. AIM w/o $W_K$)**: The ablation in 4.3 shows that removing $W_K$ from multi-head AIM causes a significant performance drop ($75.1→72.9$). Your proposed EP also lacks a $W_K$ yet achieves high performance (75.6%). Does this imply that EP's design (using full-dimensional $D_i$ queries $u_j$) successfully compensates for the lack of $W_K$, whereas simply dropping $W_K$ from a standard MHCA (like AIM) fails? A direct comparison of (AIM - $W_K$) vs. EP ($EP_{12}$) would clarify if EP's "transformation-free" design is truly superior or just more efficient.

2. **Code Implementation**:
    * **Paper (Table 1)**: The "QUERY SOURCE" for V-JEPA is listed as $q_j = W_{Q_j} u$, with $u \in \mathbb{R}^{D_i}$. This categorizes it with SimPool, where $u$ is *data-dependent* (derived from the input $X$, e.g., Global Average Pooling). This is explicitly contrasted with methods like AIM or EP, which are listed as having a learnable query $q$.
    * **Code (`poolings/jepa/attentive_pooler.py`)**: The `AttentivePooler` class (used for "jepa" in `main_linprobe.py`) implements its query as a *data-independent, learnable parameter*: `self.query_tokens = nn.Parameter(torch.zeros(1, num_queries, embed_dim))`. The paper's experimental performance for V-JEPA is based on the *code's* learnable-query architecture, but the paper's *analysis* (Table 1) attributes this performance to a *data-dependent* query architecture. Will this invalidates the paper's own framework for comparing V-JEPA against other methods like AIM or EP?

    * **Paper (Table 1 & Sec 3.3)**: The "ATTENTION" mechanism for AbMILP is explicitly defined as dot-product attention: $a = \sigma_m(K^{\top} q)$, where $K=X$ and $q$ is a single learnable query vector. Section 3.3 further claims this method "requires only $D_i$ parameters" (the size of $q$).
    * **Code (`poolings/abmilp.py`)**: The `ABMILPHead` class seemingly does *not* implement dot-product attention. Instead, it uses a multi-layer MLP (`self.attention_predictor` is `nn.Sequential`) to *predict* attention weights directly from the input features: $a = \text{softmax}(MLP(X))$. This is an MLP-based attention mechanism, not a dot-product one. The number of parameters in a two-layer MLP (such as `abmilp_depth=2`) is on the order of $D_i \times D_{hidden} + D_{hidden} \times 1$, which is far more than $D_i$ parameters (in `main_linprobe.py`, `abmilp_depth` defaults to 2). See details below:

  * Original Code (Supplementary Material `poolings/abmilp.py`)
```python
# This is the MLP-based implementation from the supplementary material.
# Note 'self.attention_predictor' is an nn.Sequential (MLP).
# This implementation's parameters are O(D^2), not O(D).
class ABMILPHead(nn.Module):
    def __init__(
            self,
            dim: int,
            activation: str= "tanh",
            depth: int = 2, # Default depth=2
            **kwargs
        ):
        super().__init__()
        self.ATTENTION_BRANCHES = 1
        attn_pred_layers = []
        for i in range(depth-1):
            attn_pred_layers.extend([
                nn.Linear(dim, dim),
                (nn.Tanh() if activation == "tanh" else nn.ReLU()),
            ])
        # The MLP: e.g., Linear(D, D) -> Tanh -> Linear(D, 1)
        attn_pred_layers.append(nn.Linear(dim, self.ATTENTION_BRANCHES))
        self.attention_predictor = nn.Sequential(*attn_pred_layers)
        # ... (other initializations) ...

    def forward(self, x: torch.Tensor) -> torch.Tensor:
        # ... (self-attn logic) ...
        predictor_input = x # (Simplified for clarity)

        # a = softmax(MLP(X))
        attn_map = self.attention_predictor(predictor_input) # [B, N, 1]
        attn_map = F.softmax(attn_map, dim=1)

        # y = V * a (where V=x)
        out = (x * attn_map).sum(dim=1)
        return out
```
   *  Based on Paper's Description, My Understanding:

```python
# This implementation strictly follows the paper's description
# (Table 1 & Sec 3.3): a = softmax(X^T q).
# It uses dot-product attention with a single learnable query 'q'.
# This implementation has O(D) parameters.
class ABMILPHead(nn.Module):
    def __init__(self, dim: int):
        super().__init__()

        # The learnable query vector 'q' (D_i parameters)
        self.query = nn.Parameter(torch.zeros(1, dim, 1))
        nn.init.xavier_uniform_(self.query)

    def forward(self, x: torch.Tensor) -> torch.Tensor:
        # x (K and V) shape: [B, N, D]

        # 1. K^T q (using batch matmul)
        # (B, N, D) @ (B, D, 1) -> (B, N, 1)
        attn_logits = x @ self.query

        # 2. a = softmax(K^T q)
        attn_map = F.softmax(attn_logits, dim=1) # [B, N, 1]

        # 3. y = V a (where V=x)
        # (B, N, D) * (B, N, 1) -> sum(dim=1) -> (B, D)
        out = (x * attn_map).sum(dim=1)
        return out
```

---

> ### Author Response · Authors · 2025-11-25
> **Answer to Reviewer 4JLQ (part 1)**
>
> We thank **Reviewer 4JLQ** for their **careful reading** of **both** our **manuscript** and **code**, and for their positive assessment of our contributions. We appreciate the reviewer’s recognition that our work provides the **first comprehensive study** of attentive probing methods, as well as their acknowledgement of the **clarity of our formulation** of Efficient Probing (EP) and the **breadth of our experimental evaluation** across diverse pre-training paradigms. We are also grateful for the reviewer’s noting that EP achieves **state-of-the-art accuracy–parameter trade-offs** and delivers substantial improvements over linear probing, particularly on models such as SimMIM and DiT where distributed representations make standard probing ineffective. We *address the reviewer’s concerns* in detail below.
>
> -----
>
> **Placement of $W_K$ ablations.**
> We thank the reviewer for this helpful suggestion. In the revised version, we have moved this analysis to the *beginning* of the analysis subsection 4.3 so that it directly supports our design choices. Furthermore, following the request of the reviewer, we also added in this paragraph the comparison between $EP_{12}$ and $AIM_{12}$ *without* $W_K$, ensuring a matched number of queries/heads.
>
> -----
>
> **Confusing terminology $\rightarrow$ "transformation-free".**
> We thank the reviewer for this helpful comment. In our original text, "transformation-free" was intended to refer specifically to the way attention weights are computed, i.e., without learnable projection matrices $W_Q$ and $W_K$. We agree that this terminology can be misleading, since the value projection $W_V$ remains an essential part of the pooling operation and is indeed important for performance. To avoid confusion, in the revised version (subsection 3.2) we have replaced the term "transformation-free" with "parameter-efficient", which more accurately reflects the design goal of our mechanism.
>
> -----

---

> > ### Author Response · Authors · 2025-11-25
> > **Answer to Reviewer 4JLQ (part 2)**
> >
> > **Matryoshka probing.**
> > We thank the reviewer for this insightful question and for highlighting the relevance of our Matryoshka analysis. We agree that the Matryoshka results in Appendix C.3, Table 11 are compelling, and they indeed show that "Efficient Matryoshka" enables a *single probing head to operate across multiple output dimensionalities without adding parameters*.
> >
> > However, our intention in this section was not to propose Matryoshka EP as the primary evaluation protocol, but rather to study whether Matryoshka representation learning—originally developed for training or fine-tuning full networks—can be meaningfully repurposed for attentive probing.
> >
> > A key point that may have caused confusion is the following:
> >
> > *Probing without Matryoshka (i.e., training independently at each output dimensionality) always provides the **upper bound** performance for that dimensionality.*
> > For example, as shown in Table 11, training EP **directly** at $D$ yields $75.6\%$, training EP **directly** at $D/2$ yields $74.4\%$, and training EP **directly** at $D/4$ yields $72.4\%$—all of which are higher than any Matryoshka-trained counterpart evaluated at these dimensions. This is expected, as independent training *specializes* the probe to a *single* dimensionality, while Matryoshka jointly optimizes *multiple* dimensionalities *at once* and therefore introduces trade-offs.
> >
> > Efficient Matryoshka EP significantly improves low-dimensional performance (e.g., +22.5% at $D/2$) while requiring no additional parameters, but its accuracy at the full dimension $D$ decreases slightly as $\lambda$ (the weight of the full-dimensional loss) decreases (e.g., $75.0\%$ at $\lambda=0.8$ and $73.9\%$ at $\lambda=0.25$ vs. the $75.6\%$ upper bound). This illustrates the inherent tension: larger $\lambda$ favors high-dimensional accuracy, whereas smaller $\lambda$ promotes better low-dimensional robustness.
> >
> > We additionally evaluate the standard linear probing (LP) baseline without Matryoshka under the same $D/2$, $D/4$, $D/8$ evaluation protocol, as also extend LP with Matryoshka. We observe that standard LP (without Matryoshka) suffers from an even sharper performance collapse than EP (e.g., 34.5% at $D/2$, 10.8% at $D/4$, and 1.4% at $D/8$). This confirms that the drop at reduced dimensions is not specific to EP, but is an **inherent limitation of evaluating ViT representations at truncated dimensionalities** **without explicit multi-scale optimization**. We have included this additional analysis in the revised version (Appendix C.3 Table 12).
> >
> > For these reasons, we view Matryoshka EP as a promising *extension* rather than the core protocol. It enables a single probe to adapt to multiple scales, but it has not yet matched the upper-bound accuracy of dimension-specific probing. A deeper study—e.g. improved multi-scale loss formulations, adaptive weighting, or structural constraints on subspaces—is needed before recommending Matryoshka EP as a general-purpose probing standard.
> >
> > We have revised the appendix text to clarify this distinction and avoid any misunderstanding. We have also included the Matryoshka LP experiments in Table 12 of the revised version. We thank the reviewer again for motivating this clarification.
> >
> > **Table: *Comparison of LP with and without Matryoshka representation learning* on ImageNet-1K (MAE ViT-B).**
> > Without Matryoshka, linear probing collapses sharply at reduced output dimensionalities. Matryoshka improves low-dimensional performance, without increasing parameters, though the accuracy at $D$ decreases as $\lambda$ (the weight of the full-dimensional loss) is reduced.
> >
> > | **Eval. on** | **LP w/o Matryoshka** | **LP w/ Matryoshka ($\lambda=0.8$)** | **LP w/ Matryoshka ($\lambda=0.6$)** | **LP w/ Matryoshka ($\lambda=0.4$)** | **LP w/ Matryoshka ($\lambda=0.25$)** |
> > |-------------|------------------------|---------------------------------------|---------------------------------------|----------------------------------------|-----------------------------------------|
> > | $D$         | 66.4                   | 66.1                                  | 65.8                                  | 65.4                                   | 65.0                                    |
> > | $D/2$       | 34.5                   | 59.2                                  | 60.6                                  | 61.2                                   | 61.4                                    |
> > | $D/4$       | 10.8                   | 52.5                                  | 54.5                                  | 55.3                                   | 55.7                                    |
> > | $D/8$       | 1.4                    | 43.4                                  | 46.1                                  | 47.0                                   | 47.4                                    |
> > -----

---

> ### Author Response · Authors · 2025-11-25
> **Answer to Reviewer 4JLQ (part 3 - last part)**
>
> **EP (12 queries) vs. AIM (12 heads) with without $W_K$.**
>
> We appreciate this insightful question. Indeed, simply removing $W_K$ from a standard multi-head cross-attention (MHCA) module such as AIM leads to a clear degradation: in our ablation of subsection 4.3, AIM$_{12}$ drops from 75.1% to 72.9% top-1 when $W_K$ is replaced by the identity. This happens because, in MHCA, each head only sees a $D_i/M$-dimensional subspace, so removing $W_K$ severely limits how each query can interact with the full token space.
>
> In contrast, EP is not "AIM without $W_K$", but a different parametrization: we use $M$ full-dimensional queries $\in \mathbb{R}^{D_i}$ (MQCA), so every query attends to the entire feature matrix and can effectively absorb the role of $W_K$. To make this explicit, we now report in subsection 4.3 a direct comparison of the three settings:
>
> | **Method**              | **Top-1 Acc.** | **# Params. (M)** |
> |-------------------------|----------------|--------------------|
> | AIM                     | 75.1           | 1.95               |
> | AIM–$W_K$               | 72.9           | 1.36               |
> | **EP ($M=12$ queries)** | **75.1**       | 1.38           |
>
> Thus, simply dropping $W_K$ from AIM fails, but our EP (12 queries) matches the full AIM (12 heads) baseline while using substantially fewer parameters. This shows that EP’s "transformation-free" (now renamed "parameter-efficient") design is not just a cheaper variant of MHCA w/o $W_K$, but **a reparameterization that preserves the expressiveness of MHCA in a more parameter-efficient way***.
>
> We have revised the phrasing of subsection 3.2 to better reflect this.
>
> -----
>
> **Code implementation (query source for V-JEPA).**
>
> Thank you for the observation. There is no mismatch between Table 1 and the code. In Table 1, V-JEPA is classified as having a *learnable* query because its query vector $u$ is a trainable parameter $\mathbf{u}\in\mathbb{R}^{D_i}$. This is why it is shown in **blue**.
>
> The code implements exactly this design: `self.query_tokens = nn.Parameter(...)`.
>
> This is in contrast to SimPool, where the query is explicitly data-dependent ($\mathbf{u} = \tfrac{1}{N}X^\top\mathbf{1}$).
>
> All V-JEPA results in the paper use the learnable-query version, so the comparison remains valid.
>
> -----
>
> **Code implementation (AbMILP).**
>
> Thank you for raising this point. The apparent discrepancy comes from the `abmilp_depth` hyperparameter. In our experiments we *explicitly set* `--abmilp_depth=1`, which reduces AbMILP to the *exact* dot-product formulation described in Table 1 and subsection 3.3:
>
> $$
> a = \mathrm{softmax}(X^\top q), \qquad q \in \mathbb{R}^{D_i},
> $$
>
> i.e. AbMILP uses a *single learnable query vector* and therefore contains only $\mathcal{O}(D_i)$ trainable parameters. When `depth=1`, the `nn.Sequential` in the code degenerates to a single linear layer $\mathbb{R}^{D_i}\to\mathbb{R}$, which is functionally identical to $X^\top q$. In this setting the parameter count of AbMILP is:
>
> $$
> D_i + 1 = 768 + 1,
> $$
>
> which is precisely what Table 1 refers to.
>
> For completeness, we also evaluate AbMILP with larger depths (`depth=2,3`), which correspond to the MLP-based variants and scale as $\mathcal{O}(D_i^2)$. Their parameter–accuracy tradeoffs are:
>
> **Table: *AbMILP architecture ablation* increasing MLP depth.**
> *# Par.: number of parameters.*
>
> | **Method**          | **Params**  | **Top-1 Acc.** |
> |---------------------|------------:|---------------:|
> | AbMILP (depth 1)    | 769,769     | 71.74          |
> | AbMILP (depth 2)    | 1,360,361   | 72.25          |
> | AbMILP (depth 3)    | 1,950,953   | 72.84          |
>
> Although deeper MLPs introduce substantially more parameters ($+\mathcal{O}(D_i^2)$), the gains in accuracy are marginal. To ensure a fair comparison in the *accuracy-versus-parameter-efficiency* setting of this paper, we used the most competitive AbMILP variant (`depth=1`) whose behaviour matches the formulation in Table 1 and subsection 3.3.
>
> We have included an explicit clarification in the revised version to avoid any confusion (subsection C.1).
>
> Finally, in response to the other reviewers’ requests regarding the relationship between PEFT and attentive probing, our revised version includes an **extended analysis in subsection 4.2**, which we believe adds *meaningful value to the community*. We would kindly invite the reviewer to consult this new material.

---

> ### Comment · Reviewer_4JLQ · 2025-11-26
>
> Thank you to the author for the detailed response and clarification; my questions and details regarding the code implementation were well addressed. I will maintain this weakly acceptance score for now. I have no further questions at this point. I will continue to follow the Q&A between the author and other reviewers and reserve the option to further increase the score to 8.

---

> > ### Author Response · Authors · 2025-12-03
> > **Answer to Reviewer 4JLQ**
> >
> > We sincerely thank the reviewer for the careful reading of our paper and the positive follow-up. We are glad that the clarifications and implementation details addressed your concerns, and we appreciate your willingness to consider raising your score to 8.

---

### Author Response · Authors · 2025-12-04
**Summary**

**Paper in one line**

Attentive probing is the right protocol for patch-token models; EP is a simple multi-query cross-attention head that beats linear probing and prior attentive probes while being more parameter- and compute-efficient.

---
**Key rebuttal contributions**

- **PEFT vs EP comparison (Sec. 4.2, Fig. 4 + new tables)**
  Added a large-scale study on MAE ViT-B/16 comparing:
  - Baselines: [CLS], AbMILP, DELF, AIM
  - **EP at multiple output dims**
  - **>40 LoRA variants** (single-layer / all-layer, different matrices and ranks)
  - **BitFit and LayerNorm tuning**
  → EP lies on / above the PEFT Pareto front for similar parameter budgets.

- **Hybrid PEFT + EP (LoRA+EP, BitFit+EP)**
  Introduced and evaluated hybrid configurations (e.g. LoRA\_all + EP, BitFit + EP).
  → These hybrids form a new dominant region in the accuracy–parameter plane, strictly improving over both pure EP and pure PEFT.

- **Cross-dataset k-NN study (EP vs LoRA)**
  New k-NN experiments on MAE ViT-B/16 using ImageNet-1K, Cars196, Food-101:
  - Compare frozen backbone vs EP-probed vs best LoRA-tuned features.
  → LoRA best in-domain; **EP consistently stronger or comparable out-of-domain**, showing complementary behavior.

- **Zero-shot VLMs vs EP on vision encoder**
  New experiments on CLIP ViT-L/14, SigLIP ViT-L/16, SigLIP2 ViT-L/16 (ImageNet-1K):
  - Report zero-shot accuracy of full VLM vs **vision encoder + EP** (frozen backbone).
  → EP improves zero-shot by **+4.5 to +7.9 points**, showing that strong VLMs still benefit from probing.

- **EP built into CLIP pre-training (no downstream training)**
  New experiment training CLIP from scratch on YFCC-100M with EP integrated:
  - At downstream time, perform zero-shot ImageNet-1K by replacing [CLS] with EP pooled vector.
  → **33.4% → 36.7%** zero-shot top-1, demonstrating EP as a **built-in, zero-cost probe**.

- **EP for non-classification tasks: WSOL localization**
  Added unsupervised ImageNet-1K WSOL evaluation (MaxBoxAccV2) using only EP attention maps (no extra training), across 7 backbones.
  → Average **+9.8 points** over standard last-layer [CLS]→patch attention.

- **EP for non-classification tasks: image retrieval (CUB & Cars)**
  New zero-shot retrieval experiments (R@K) on CUB-200-2011 and Cars196:
  - Compare frozen backbone features vs EP features (EP trained on ImageNet-1K).
  → EP features **substantially outperform** backbone features across multiple backbones (MAE, BEiTv2, iBOT, CLIP, SigLIP).

- **EP convergence analysis (EP@10 vs LP@90)**
  New convergence table on 12 backbones (MAE, BEiTv2, DINOv2/v3, iBOT, CLIP, SigLIP, AIMv2):
  - Report EP accuracy after 10 epochs vs LP@90 and EP@90.
  → **EP@10 already matches or exceeds LP@90**, and recovers **>97% of EP@90** on average.

- **Frozen-feature probing experiment**
  New experiment on MAE ViT-B/16 with **pre-stored patch features** on ImageNet-1K:
  - Compare LP vs EP in this “frozen-feature” regime.
  → Both drop by ~2 points vs on-the-fly training, validating the setting but motivating better feature-space augmentations (flagged as future work).

- **EP vs AIM with / without W_K at matched queries/heads**
  Added explicit ablation comparing:
  - AIM (12-head MHCA), AIM with W_K removed, and EP (12-query MQCA).
  → EP (12 queries) **matches AIM (12 heads) at 75.1%** with **~30% fewer parameters**, while “AIM without W_K” degrades to 72.9%.

- **Matryoshka probing extension (LP)**
  Extended analysis of Matryoshka representation learning:
  - **New LP experiments** showing LP collapses even more severely at low dims, while Matryoshka LP improves low-dim performance without extra parameters, but at the cost of degrading the full-dimensional performance.

- **Scalability experiment on ImageNet-21K**
  New probe on MAE ViT-B/16 using ImageNet-21K (14M images, >19K classes):
  → EP continues to outperform LP (e.g., **31.9% → 35.6%**), showing robust scaling to large datasets.

- **Fast backbone-quality assessment “in the era of large models”**
  As a direct answer to industrial/large-model concerns, we integrate EP into CLIP pre-training on YFCC-100M and **use it as a built-in zero-shot head**:
  → Zero-shot ImageNet-1K improves from **33.4% (standard CLIP head)** to **36.7% (CLIP+EP)**, **without any extra downstream training**, demonstrating that EP can act as a cost-free probe during backbone training.

---

### Meta-Review · Area_Chair_gYHB · 2026-01-07

**Summary:**

This paper proposes "Efficient Probing" (EP), a lightweight attention-based probing method. Overall, the proposed method is practically useful for evaluating MIM models efficiently. However, several critical concerns were raised:

**Technical Novelty and Significance:** The most significant and recurring concern was the perceived lack of technical novelty. Reviewers viewed the removal of projection matrices ($W_k, W_v$) from standard cross-attention as a technically minor or "incremental" modification.

**Baselines and Benchmarking:** Reviewers noted gaps in the empirical evaluation, specifically requesting broader comparisons with existing attentive probing methods and stronger baselines to validate the claim that EP "consistently outperforms" prior approaches. There were concerns about whether the "benchmark-reality gap" was fully addressed.

**Reviewer Concerns:**

The rebuttal effectively reinforced the "lens of efficiency" argument and demonstrated that EP achieves a better Pareto frontier. The authors also clarified that the simplicity of EP is a design feature intended to remove redundancy in probing setups. The issue of novelty remains for reviewers looking for deeper theoretical insights.

**Reviewer Scores:**

Reviewer Vb6E, who was the most critical regarding novelty, decided to raise the score from 2 to 4, and Reviewers 4JLQ and Ph7P would likely maintain their scores at 6.

---

### Decision · Program_Chairs · 2026-01-26

Accept (Poster)